# Structural basis for APE1 processing DNA damage in the nucleosome

Tyler M. Weaver[1,2], Nicole M. Hoitsma[1], Jonah J. Spencer[1], Lokesh Gakhar [3,4,5], Nicholas J. Schnicker [4] & Bret D. Freudenthal [1,2,6] ✉

Genomic DNA is continually exposed to endogenous and exogenous factors that promote DNA damage. Eukaryotic genomic DNA is packaged into nucleosomes, which present a barrier to accessing and effectively repairing DNA damage. The mechanisms by which DNA repair proteins overcome this barrier to repair DNA damage in the nucleosome and protect genomic stability is unknown. Here, we determine how the base excision repair (BER) endonuclease AP-endonuclease 1 (APE1) recognizes and cleaves DNA damage in the nucleosome. Kinetic assays determine that APE1 cleaves solvent-exposed AP sites in the nucleosome with 3 − 6 orders of magnitude higher efficiency than occluded AP sites. A cryo-electron microscopy structure of APE1 bound to a nucleosome containing a solvent-exposed AP site reveal that APE1 uses a DNA sculpting mechanism for AP site recognition, where APE1 bends the nucleosomal DNA to access the AP site. Notably, additional biochemical and structural characterization of occluded AP sites identify contacts between the nucleosomal DNA and histone octamer that prevent efficient processing of the AP site by APE1. These findings provide a rationale for the position-dependent activity of BER proteins in the nucleosome and suggests the ability of BER proteins to sculpt nucleosomal DNA drives efficient BER in chromatin.

Genomic DNA of eukaryotic cells is packaged into chromatin through a fundamental repeating unit called the nucleosome. The nucleosome consists of ~147 bp of DNA wrapped around a core octameric histone protein complex, containing two copies each of histone H2A, H2B, H3, and H4[1]. The compact structure of the nucleosome and the robust contacts of the histone octamer with the nucleosomal DNA generates a barrier to accessing the genomic DNA sequence. Importantly, this nucleosome barrier must be overcome during critical cellular processes such as transcription, DNA replication, and DNA repair.

Both chromatinized and non-nucleosomal DNA are susceptible to endogenous and exogenous sources of DNA damage. One of the most common forms of DNA damage is the baseless sugar moiety known as apurinic/apyrimidinic (AP) sites, with an estimated ~10,000 AP sites

generated in each cell per day[2,3]. These AP sites are generated through spontaneous depurination and depyrimidination, as well as through the excision of damaged DNA bases by damage-specific DNA glycosylases. Repair of AP sites is critical for maintaining genomic stability[4] as they lack coding information during DNA replication[5], are prone to generating cytotoxic DNA breaks[6], and form DNA-protein crosslinks (DPCs) in the nucleosome[7,8]. The primary enzyme tasked with locating and processing genomic AP sites is the multifunctional nuclease AP endonuclease 1 (APE1), which is a key component of the base excision repair (BER) pathway[9]. During BER, APE1 functions as an AP-endonuclease by cleaving the DNA phosphodiester backbone 5′ of the AP site generating a 5′-nicked BER intermediate. The 5′-nicked BER intermediate is then processed by additional BER proteins, which

[1]Department of Biochemistry and Molecular Biology, University of Kansas Medical Center, Kansas City, KS 66160, USA. [2]Department of Cancer Biology, University of Kansas Medical Center, Kansas City, KS 66160, USA. [3]Department of Biochemistry and Molecular Biology, University of Iowa Carver College of Medicine, Iowa City, IA 52242, USA. [4]Protein and Crystallography Facility, University of Iowa Carver College of Medicine, Iowa City, IA 52242, USA. [5]Department of Biology, Massachusetts Institute of Technology, Cambridge, MA 02142, USA. [6]University of Kansas Cancer Center, Kansas City, KS 66160, USA. ✉e-mail: bfreudenthal@kumc.edu

ultimately restore the coding potential of the DNA to maintain genome stability. The biological importance of APE1, and the processing of genomic AP sites, is highlighted by the embryonic lethality of APE1 knockout in *M. musculus*[10], and the sensitivity of APE1-deficient cells to DNA damaging agents[11,12].

The mechanisms used by APE1 and other BER enzymes to repair DNA damage in non-nucleosomal DNA are well-established[9,13,14]. However, how these enzymes function in the context of nucleosomes and higher-order chromatin remains poorly understood. In-vitro studies using recombinant nucleosomes have shown the activity of APE1 is highly dependent on the position of the DNA damage (i.e., the AP site) in the nucleosome, where solvent-exposed AP sites are more readily processed than occluded AP sites[15–18]. This position-dependent activity in the nucleosome is not exclusive to APE1 and is shared among other proteins in the BER pathway[18–27]. Consistent with these in-vitro observations, genome-wide repair of nucleosomal DNA base damage by BER proteins is also dependent on the position of the DNA damage in the nucleosome[28–30]. To date, the mechanisms BER proteins use for recognition and processing of DNA damage in the nucleosome, and the molecular basis for the position-dependent DNA repair activity remains unknown. Here, we utilize biochemical assays and cryo-electron microscopy (cryo-EM) to determine the mechanism used by APE1 to process nucleosomal AP sites and provide mechanistic insight into the position-dependent repair of nucleosomal DNA damage.

## Results

### APE1 AP-endonuclease activity is position-dependent in the context of the NCP

To investigate how the nucleosome impacts the AP-endonuclease activity of APE1, we generated three recombinant NCPs that each contain a single tetrahydrofuran AP site analog. These three AP sites are positioned at distinct locations within the NCP described in terms of the position of the AP site relative to the nucleosome dyad, or superhelical location (SHL) (Fig, 1a and Supplementary Fig. 1a). Moreover, the AP sites also represent different rotational orientations, which dictates whether the phosphate backbone of the AP site is facing outwards away from the histone octamer (solvent-exposed) or facing inward towards the histone octamer (occluded). One nucleosome contains an AP site in a solvent-exposed rotational orientation near the nucleosome entry/exit site at $SHL_{-6}$, referred to as $AP\text{-}NCP_{-6}$. The second nucleosome contains an occluded AP site near the nucleosome entry/exit site at $SHL_{-6.5}$, referred to as $AP\text{-}NCP_{-6.5}$. The third nucleosome contains an occluded AP site one bp adjacent to the nucleosome dyad at $SHL_0$, referred to as $AP\text{-}NCP_0$.

To quantitatively characterize the position-dependent activity of APE1 on AP-NCPs, we performed single-turnover pre-steady state enzyme kinetics on $AP\text{-}NCP_{-6}$, $AP\text{-}NCP_{-6.5}$, and $AP\text{-}NCP_0$ substrates (Fig. 1b and Supplementary Fig. 1c–e). Under single-turnover conditions, where APE1 is in excess of the AP-NCP substrate, the kinetic time course corresponds to the first enzymatic turnover and can be fit to determine the APE1 DNA cleavage rate ($k_{obs}$). APE1 exhibits biphasic kinetics with each of the AP-NCPs, which indicates the presence of two distinct cleavage rates (Fig. 1c and Supplementary Fig. 1b). Here, we discuss the cleavage rate determined for the major population for each AP-NCP substrate as this likely represents the biologically relevant rate for AP-endonuclease activity. For $AP\text{-}NCP_{-6}$, the observed rate constant for APE1 was $500 \pm 40\,s^{-1}$, which is similar to the $k_{obs}$ of $441 \pm 40\,s^{-1}$ for APE1 on non-nucleosomal duplex DNA[31]. When the AP site was placed in an occluded position near the nucleosome entry/exit site, $AP\text{-}NCP_{-6.5}$, we observed a decrease in APE1 cleavage rate compared to the solvent-exposed $AP\text{-}NCP_{-6}$. The APE1 cleavage rate for $AP\text{-}NCP_{-6.5}$ was $0.12 \pm 0.01\,s^{-1}$, which is a 3,650-fold decrease compared to non-nucleosomal duplex DNA. The largest reduction in APE1 activity was observed for the occluded AP site near the nucleosome dyad, $AP\text{-}NCP_0$. The APE1 cleavage rate for $AP\text{-}NCP_0$ was $1.30\times10^{-4} \pm 0.23\times10^{-4}\,s^{-1}$,

which is a 3,400,000-fold reduction compared to non-nucleosomal duplex DNA. To determine the specificity of APE1 cleavage for the nucleosomal AP sites, we generated a non-damaged nucleosome (ND-NCP) that lacks an AP site (Supplementary Fig. 1f). Importantly, APE1 cleavage activity was not observed for ND-NCPs, confirming the cleavage specificity of APE1 for nucleosomal AP sites (Supplementary Fig. 1g). Together, these pre-steady state kinetic measurements indicate that APE1 rapidly cleaves solvent-exposed AP sites at the nucleosome entry/exit site ($AP\text{-}NCP_{-6}$). In contrast, occluded AP sites at the nucleosome entry/exit ($AP\text{-}NCP_{-6.5}$) and dyad ($AP\text{-}NCP_0$) are cleaved with moderate and low efficiency, respectively.

To determine if the observed differences in APE1 cleavage observed at the three AP site positions is the result of reduced substrate binding, we performed electrophoretic mobility shift assays (EMSAs) with APE1 and ND-NCP, $AP\text{-}NCP_{-6}$, $AP\text{-}NCP_{-6.5}$, and $AP\text{-}NCP_0$ (Fig. 1d and Supplementary Fig. 1h-k). EMSAs with ND-NCPs identified APE1 robustly interacts with the NCP ($K_{d,app}$ of $71 \pm 10$ nM) even in the absence of an AP-site (Supplementary Fig. 1k,l). Analysis of the EMSAs for AP-NCPs yielded $K_{d,app}$ of $17 \pm 3$ nM for $AP\text{-}NCP_{-6}$, $20 \pm 7$ nM for $AP\text{-}NCP_{-6.5}$, and $20 \pm 8$ nM for $AP\text{-}NCP_0$, a modest ~3.5 fold higher affinity than ND-NCPs (Fig. 1e). Importantly, the similar affinities of APE1 for all three AP-NCPs indicates the differences in catalytic efficiencies for $AP\text{-}NCP_{-6}$, $AP\text{-}NCP_{-6.5}$, and $AP\text{-}NCP_0$ are not the result of reduced nucleosome binding by APE1. However, we cannot rule out subtle differences in specific binding of APE1 to each individual AP-site within the nucleosome due to the non-specific binding observed between APE1 and non-damaged nucleosomal DNA in our EMSA assays.

### Mechanism for APE1 processing solvent-exposed AP sites in the nucleosome

The pre-steady state kinetics measurements indicate that APE1 rapidly processes a solvent-exposed AP site at $SHL_{-6}$. To obtain mechanistic insight into how APE1 efficiently binds and processes the solvent-exposed AP site at $SHL_{-6}$, we generated and purified an $APE1\text{-}AP\text{-}NCP_{-6}$ substrate complex for structure determination by cryo-EM (Supplementary Fig. 2a). A subset of 58,854 particles was used to generate a 3.4 Å cryo-EM reconstruction of the $APE1\text{-}AP\text{-}NCP_{-6}$ complex (Fig. 2, Supplementary Fig. 2, Supplementary Table 1). Both APE1 and the $AP\text{-}NCP_{-6}$ are well-resolved in the cryo-EM map, with local resolutions of ~4.5–6 Å for APE1 and ~3–4 Å for $AP\text{-}NCP_{-6}$ (Supplementary Fig. 2c, d). Interestingly, only the catalytic core of APE1 (residues 43–318) was observed in the $APE1\text{-}AP\text{-}NCP_{-6}$ reconstruction despite utilizing full-length APE1 during cryo-EM grid preparation. The inability to resolve a large portion of the APE1 N-terminal domain (residues 1–42) in our cryo-EM reconstruction is likely due to significant conformational flexibility and suggests it does not form a stable interaction with the nucleosome when bound at this AP site location. This conformational flexibility of the N-terminal domain is consistent with prior structural studies of APE1[32].

In the $APE1\text{-}AP\text{-}NCP_{-6}$ structure, APE1 is engaged with the nucleosome at the entry/exit site with a ~10 base-pair footprint that spans the nucleosomal DNA between $SHL_{-5.5}$ to $SHL_{-6.5}$ (Fig. 3a). Importantly, APE1 only interacts with the nucleosomal DNA and no direct contacts between APE1 and the core histone octamer are observed. The interaction between APE1 and the nucleosome occurs through three distinct molecular interfaces that make extensive contacts with both the non-damaged and damaged strands of the nucleosomal DNA (Fig. 3b). The first interface consists of the positively charged APE1 side chains (R73, K78, K98, and K103) that coordinate the phosphate backbone of the non-damaged DNA strand at $SHL_{-5.5}$. (Fig. 3c). The second interface consists of a series of positively charged APE1 lysine side chains (K224, K227, K228) that are in position to interact with the phosphate backbone of the non-damaged DNA strand near $SHL_{-6.5}$ (Fig. 3d). The third interface is the APE1 active site, which

interacts extensively with the nucleosomal DNA at SHL$_{-6}$. The APE1 active site encompasses the nucleosomal DNA at SHL$_{-6}$ by wedging R177 and the intercalating loop (residues 269 - 271) into the major and minor grooves of the nucleosomal DNA, respectively (Fig. 3e). This positions the APE1 active site in a conformation to make extensive interactions with the damaged DNA strand surrounding the AP site (Fig. 3e, discussed below). Notably, the interactions observed for APE1 and the nucleosomal DNA between SHL$_{-5.5}$ to SHL$_{-6.5}$ are similar to those previously reported for non-nucleosomal DNA, indicating a common mechanism for engaging both DNA substrates[33,34].

APE1 utilizes a DNA sculpting mechanism for AP site recognition and catalysis in non-nucleosomal duplex DNA, where the DNA is bent ~35° to evict the AP site from the DNA helix into the APE1 active site[33,34].

To determine if APE1 sculpts nucleosomal DNA, we obtained a 3.0 Å cryo-EM reconstruction of the AP-NCP$_{-6}$ in the absence of APE1 (Supplementary Fig. 3 and Supplementary Table 1). Comparison of the APE1-AP-NCP$_{-6}$ and AP-NCP$_{-6}$ structures revealed significant distortion of the nucleosomal DNA upon APE1 binding (Fig. 4a). APE1 binding induces an additional ~20° bend in the nucleosomal DNA from SHL$_{-5.5}$ to SHL$_{-6.5}$ that results from a ~7 Å movement of the nucleosomal DNA away from the histone octamer at SHL$_{-6}$. In addition, APE1 binding results in significant widening of the minor groove of the nucleosomal DNA containing the AP site. The APE1-induced DNA bending does not cause significant structural rearrangements of the histone octamer, and contacts made between the histone octamer and the nucleosomal DNA at SHL$_{-5.5}$ and SHL$_{-6.5}$ remain intact (Supplementary Fig. 4a−c).

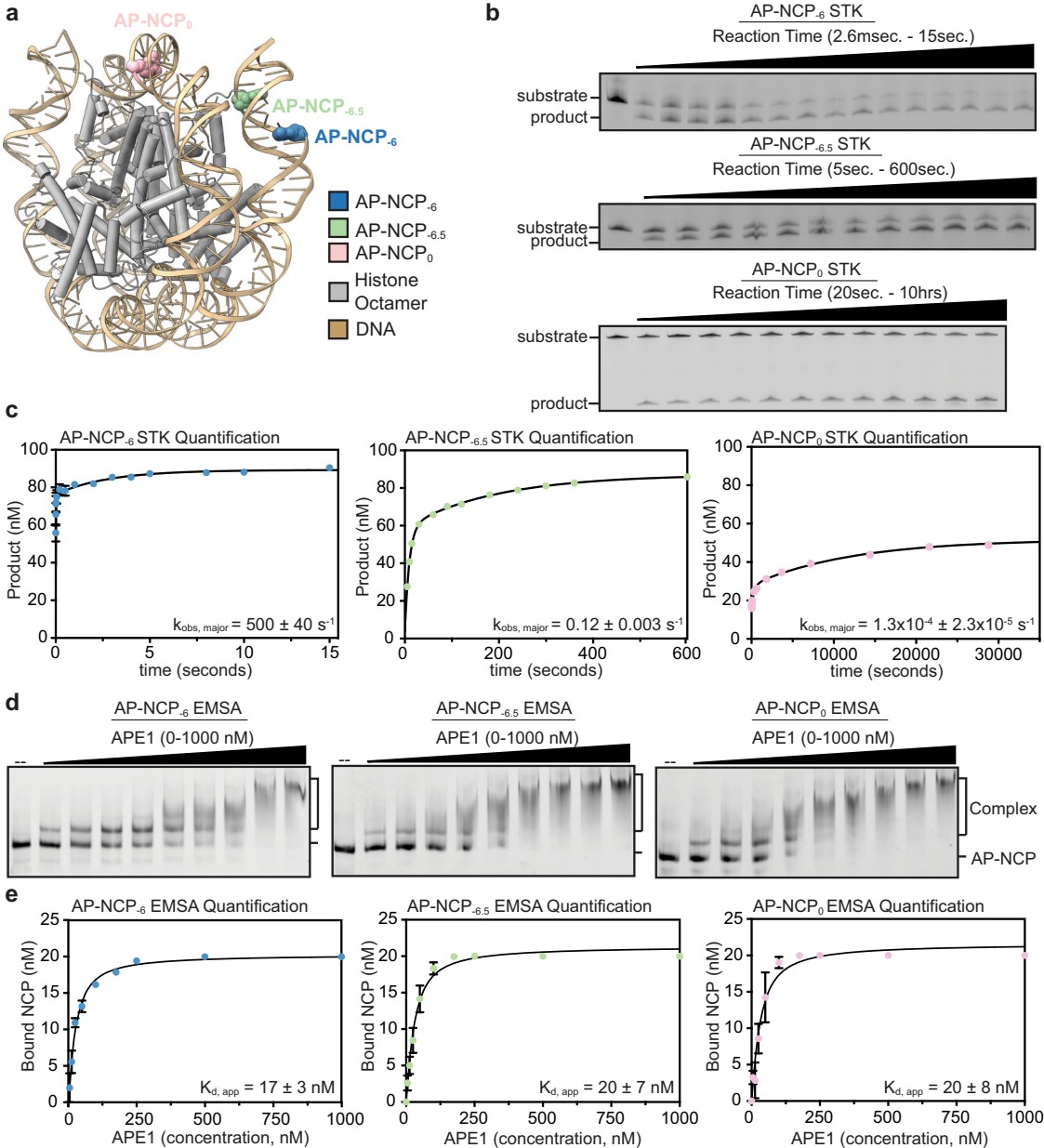

**Fig. 1 | APE1 endonuclease activity is position-dependent in the nucleosome. a** Structure of a nucleosome core particle (PDB:4JJN) with the AP site positions at SHL$_{-6}$, SHL$_{-6.5}$, and SHL$_0$ labeled. **b** Representative gels for APE1 single turnover kinetic (STK) experiments with substrate and product bands detected using the 6-FAM label. **c** Quantification with fits for the AP-NCP$_{-6}$, AP-NCP$_{-6.5}$, and AP-NCP$_0$ single turnover kinetic (STK) experiments. The data shown are the mean ± standard error of the mean from three replicate experiments. The error bars are included but are smaller than the circles used to represent the data points **d** Representative gels for the APE1 EMSA experiments with the free AP-NCP and APE1-bound AP-NCP (complex) bands detected using the 6-FAM label. **e** Quantification with fits for the AP-NCP$_{-6}$, AP-NCP$_{-6.5}$, and AP-NCP$_0$ EMSA experiments. The data shown are the mean ± standard deviation from three replicate experiments. See Supplementary Fig. 1 for associated data. Source data for this figure are provided as a Source Data file.

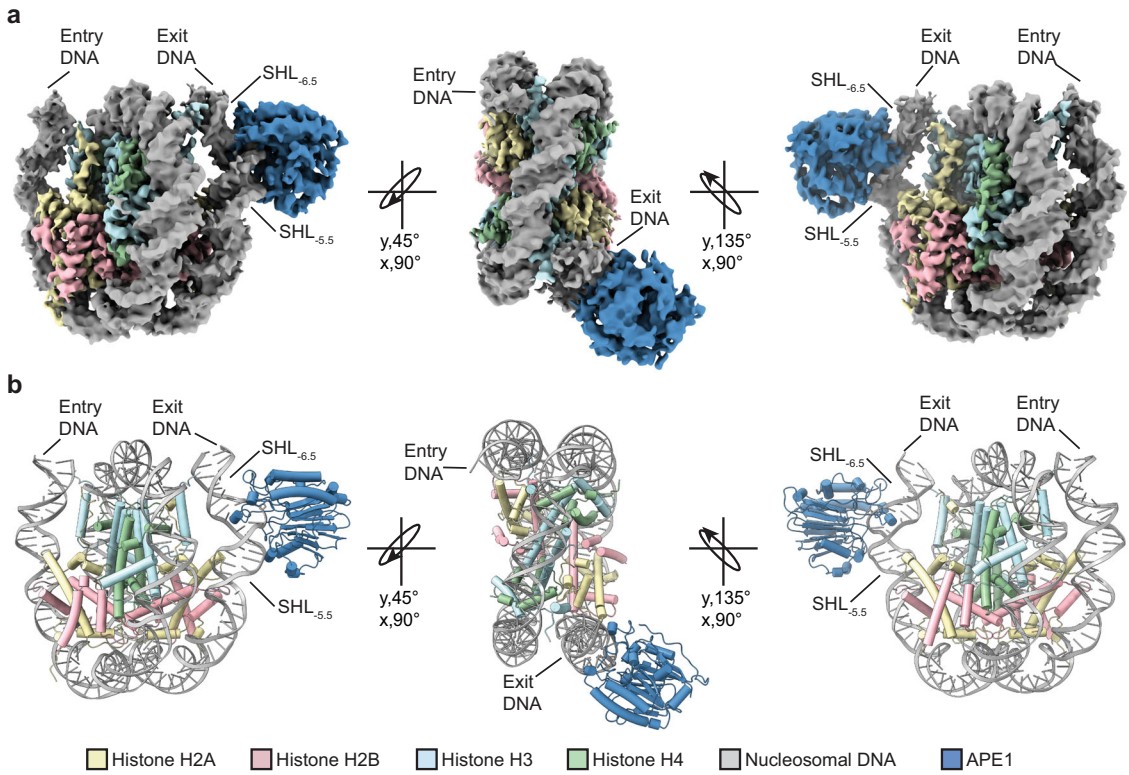

**Fig. 2 | Cryo-EM structure of the APE1-AP-NCP$_{-6}$ complex. a** A sharpened cryo-EM map of the APE1-AP-NCP complex in three different views. **b** Model of the APE1-AP-NCP$_{-6}$ complex shown in the same three orientations as (a).

Histone H2A ☐ Histone H2B ☐ Histone H3 ☐ Histone H4 ☐ Nucleosomal DNA ☐ APE1 ☐

In addition to DNA bending, significant variability in the APE1 position relative to the histone octamer was observed during 3D classification of the cryo-EM data (Supplementary Fig. 2b). Subsequent 3D variability analysis of the APE1-AP-NCP$_{-6}$ particles revealed a substantial translational movement of APE1 around the nucleosomal DNA that is centered on the AP site, suggesting conformational heterogeneity during AP-site recognition in the nucleosome (Supplementary Fig. 4d).

The APE1-induced DNA bending facilitates displacement of the AP site from within the nucleosomal DNA helix into the APE1 active site (Fig. 4b). Importantly, this extrahelical conformation of the AP site is different than the predominantly intrahelical conformation seen in the AP-NCP$_{-6}$ structure (Supplementary Fig. 4e). The void in the DNA helix generated by the extrahelical AP site is filled by R177, which wedges into the major groove of the nucleosomal DNA near SHL$_{-6}$ (Fig. 4c). In this conformation, R177 sits across from the orphan DNA base of the non-damaged DNA strand and likely stabilizes the extrahelical conformation of the AP site. The APE1 active site makes additional contacts with the nucleosomal DNA at SHL$_{-6}$ through intercalating loop residues Y269, M270, and M271, which wedge into the minor groove of the nucleosomal DNA (Fig. 4d). While several APE1 side chains were readily observed in the cryo-EM map, we did not observe clear density for all the side chains within the APE1 active site involved in catalysis. The remaining side-chain conformations for APE1 catalytic residues were modeled using high-resolution X-ray crystal structures of APE1[33,34]. Importantly, the APE1 side chains that coordinate the nucleophilic water (residues D210 and N212) and catalytic metal (residues D70, E96, and D308) are in position to perform catalysis on the nucleosomal AP site (Fig. 4e), suggesting our structure represents APE1 in a catalytically competent conformation. Additional structural comparison of our APE1-AP-NCP$_{-6}$ structure with a high-resolution crystal structure of APE1 bound to non-nucleosomal duplex DNA containing an AP site (APE1-AP-DNA, PDB:5DFI) revealed a similar mode of interaction with both substrates (Supplementary Fig. 4f)[33,34].

To obtain additional insight into the catalytic mechanism APE1 uses to cleave solvent-exposed AP sites in the nucleosome, we generated three APE1 active site mutations (E96Q/D210N, R177A, and Y269A). In the APE1 AP-endonucleolytic cleavage reaction, residues E96 and D210 are critical for coordinating the catalytical metal and nucleophilic water[35–37], respectively, R177 can act as a surrogate base to stabilize the extrahelical conformation of the AP site[33,38], and Y269 is involved in APE1-mediated DNA sculpting[39]. We used a product formation assay to determine how well each APE1 mutant performs cleavage of AP-NCP$_{-6}$ (Fig. 4f and Supplementary Fig. 5a). The APE1 E96Q/D210N mutant did not result in appreciable product formation for AP-NCP$_{-6}$, suggesting this mutant is catalytically dead. The APE1 R177A mutant resulted in a significant decrease in product formation for AP-NCP$_{-6}$, whereas Y269A had a minimal effect on product formation for AP-NCP$_{-6}$ compared to WT APE1. This suggests that coordination of the catalytic metal and nucleophilic water, as well as stabilization of the extrahelical AP site conformation are important for APE1 cleavage of nucleosomal AP sites. Importantly, the E96Q/D210N, R177A, and Y269A mutants all maintain the ability to bind AP-NCP$_{-6}$, though subtle differences in binding pattern were observed (Supplementary Fig. 5b). This indicates the reduction in product formation is not due to an overall reduction in APE1 nucleosome binding.

## Mechanism for APE1 processing occluded AP sites in the nucleosome

To further understand the mechanism APE1 uses to bind and cleave AP-NCP$_{-6.5}$ and AP-NCP$_0$, we attempted to determine structures of APE1 bound to AP-NCP$_{-6.5}$ and AP-NCP$_0$. Despite multiple attempts, we were unable to obtain a reconstruction of APE1 bound to either AP-NCP$_{-6.5}$ or AP-NCP$_0$ (Supplementary Fig. 6a and 7a, see "Methods" section). The exact reason for this is unclear, but likely indicates a high level of heterogeneity in APE1 binding position for AP-NCP$_{-6.5}$ and AP-NCP$_0$. To obtain additional insight into the catalytic mechanism APE1 uses to

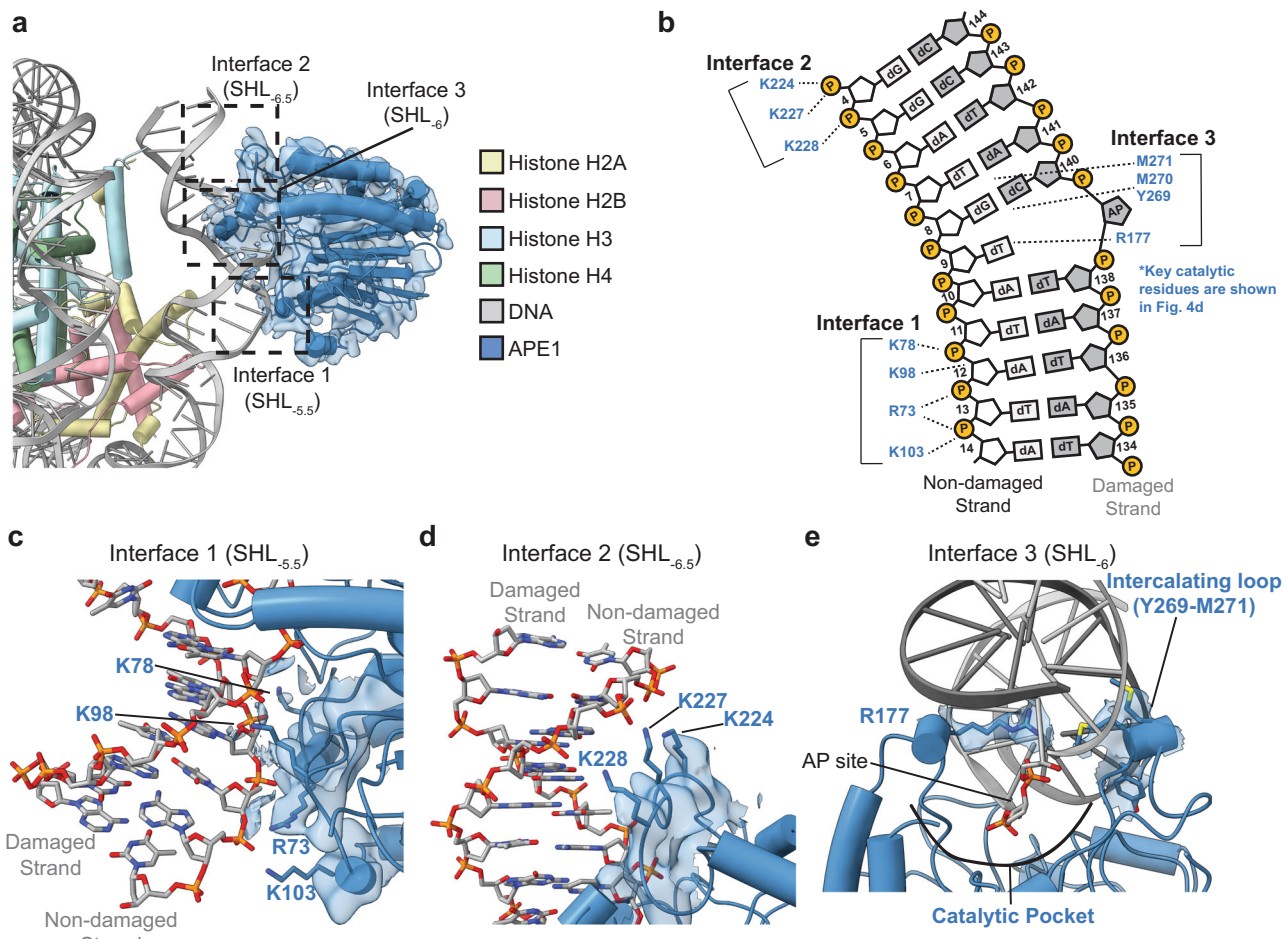

**Fig. 3 | APE1 interacts with AP-NCP-6 using three molecular interfaces. a** A focused view of the APE1-AP-NCP-6 complex. Black dotted squares denote the three distinct molecular interfaces. The segmented APE1 cryo-EM map is shown as a transparent blue surface. **b** A diagram representing the contacts between APE1 and the nucleosomal DNA identified using PLIP[66]. Several APE1 catalytic residues are not shown for clarity (see Fig. 4e). **c** Focused view of

interface 1 at SHL−5.5 in the APE1-AP-NCP-6 complex. **d** Focused view of interface 2 at SHL−6.5 in the APE1-AP-NCP−6 complex. **e** Focused view of interface 3 at SHL−6 in the APE1-AP-NCP−6 complex. The cryo-EM map is shown as a transparent blue surface in (**c−e**). Key APE1 active site residues are colored blue, but sticks representing some APE1 catalytic side chains are omitted for clarity in (**e**).

cleave APE1-AP-NCP-6.5 and APE1-AP-NCP0, we utilized the product formation assay to test the cleavage ability of the APE1 active site mutants (E98A/D210N, R177A, and Y269A) for AP-NCP-6.5 and AP-NCP0 (Fig. 5a and Supplementary Fig. 5c,e). The E96Q/D210N mutation completely abrogated APE1 endonuclease activity for AP-NCP-6.5 and AP-NCP0, whereas the APE1 R177A and Y269A mutations resulted in a moderate decrease in product formation for AP-NCP-6.5 and AP-NCP0. Importantly, the differences in product formation for AP-NCP-6.5 and AP-NCP0 are not due to reduced nucleosome binding, though subtle changes in binding patterns were observed (Supplementary Fig. 5d, f). The overall trends in the product formation assay for APE1 E96Q/ D210A and R177A mutants cleaving AP-NCP-6.5 and AP-NCP0 are consistent with what was observed for AP-NCP-6, suggesting APE1 uses the same general catalytic mechanism for cleavage of all three AP-NCPs (compare Figs. 5a and 4f). However, the Y269A mutant had a small but significant decrease in product formation for AP-NCP-6.5 and AP-NCP0, which was not observed for the solvent-exposed AP-NCP-6.

The product formation assays indicate APE1 uses a similar catalytic mechanism for cleaving all three AP-NCPs. This suggests the differences in APE1 cleavage rate for each AP-NCP may result from the inability of APE1 to access AP sites in the nucleosome. To determine any structural differences between the three AP-NCPs, we obtained two additional cryo-EM reconstructions of AP-NCP-6.5 and AP-NCP0 at 3.4 Å and 4.0 Å, respectively (Supplementary Fig. 6, 7 and

Supplementary Table 1). Structural comparison of the AP-NCP-6, AP-NCP-6.5, and AP-NCP0 revealed minimal changes to the overall structure of the nucleosome, indicating the three different AP sites do not induce large-scale conformational changes in the nucleosome (Supplementary Fig. 8a). In all three AP-NCP structures the AP site adopts a major conformation that is intrahelical, though we cannot rule out heterogeneity in the AP site conformation (Supplementary Fig. 8b). Closer inspection of three AP-NCP structures revealed significant differences in how extensive the nucleosomal DNA surrounding the AP sites interacts with the core histone octamer. The AP site at SHL−6 is completely solvent-exposed and no direct contacts between the histone octamer and damaged strand of the nucleosomal DNA are observed (Fig. 5b). In contrast, the damaged DNA strand containing the AP sites at SHL−6.5 and SHL0 make extensive interaction with the histone octamer. The AP site and surrounding nucleosomal DNA at SHL−6.5 interacts with the αN-helix of histone H3 and is in close proximity to the C-terminal tail of histone H2A (Fig. 5c). Similarly, the nucleosomal DNA surrounding the AP site at SHL0 is adjacent to the H3-H3′ interface at the nucleosome dyad (Fig. 5d). Subsequent modeling of APE1 bound at these occluded AP site positions revealed significant clashes between APE1 and the histone octamer that are incompatible with APE1 binding and DNA sculpting (Supplementary Fig. 8c). Together, this indicates large-scale structural rearrangements in the nucleosomal DNA and/or histone octamer are needed for processing occluded AP

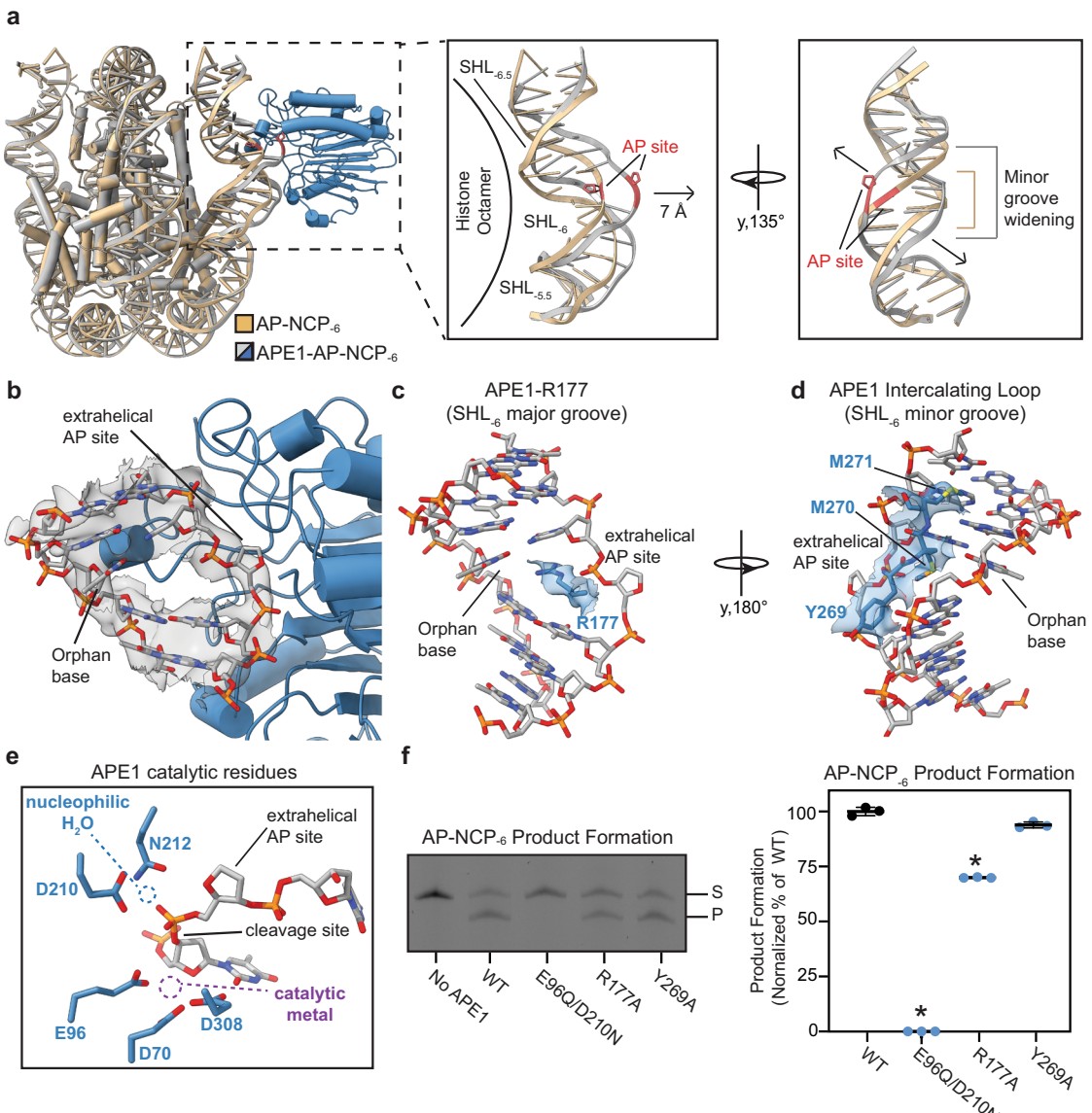

**Fig. 4 | Mechanism of nucleosomal AP site recognition by APE1. a** Structural overlay of the APE1-AP-NCP$_{-6}$ complex and AP-NCP$_{-6}$. Focused view of the nucleosomal DNA between SHL$_{-5.5}$ and SHL$_{-6.5}$ without APE1 shown highlighting changed in the nucleosomal DNA upon APE1 binding. **b** Focused nucleosomal DNA view of the AP site in the APE1 active site. The cryo-EM map is shown as a transparent gray surface. **c** Focused view of R177 in the APE1 active site at SHL$_{-6}$. **d** Focused view of the APE1 intercalating loop at SHL$_{-6}$. The cryo-EM map is shown as a transparent blue surface for (**c** and **d**). **e** Focused view of and the nucleophilic water and metal coordinating residues in the APE1 active site. The nucleophilic water and metal locations from PDB:5DFI are labeled. **f** Representative gel of the AP-NCP$_{-6}$ product formation assays for WT, E96Q/D210N, R177A, and Y269A APE1 (left). The substrate (S) and product (P) bands were detected using the 6-FAM label. Quantification of the AP-NCP$_{-6}$ product formation assays for WT, E96Q/D210N, R177A, and Y269A APE1 (right). The data shown are the mean ± standard deviation from the three replicate experiments. The * denotes values that are significantly different ($p < 0.01$) than WT APE1 as determined by two-tailed student t test. P values: WT vs E96Q/D210N ($p$-0.0001), WT vs R177A ($p$-0.0011). Source data for this figure are provided as a Source Data file.

sites, which explains the moderate and low APE1 cleavage rates for AP-NCP$_{-6.5}$ and AP-NCP$_0$ (Fig. 1c and Supplementary Fig. 1b).

## Discussion

The cellular repair of AP sites is critical for maintaining genomic stability[4–6]. Our work describes how the essential BER protein APE1 recognizes and cleaves AP-sites in the nucleosome, providing a foundation for understanding how DNA repair occurs in chromatin. Our kinetic analysis revealed APE1 rapidly cleaves a solvent-exposed AP site near SHL$_{-6}$, with a cleavage rate similar to non-nucleosomal DNA. The cryo-EM structure of the APE1- AP-NCP$_{-6}$ complex provides a mechanistic basis for this observation. Nucleosomal AP site recognition by APE1 is accomplished through a DNA sculpting mechanism, where APE1 bends the DNA and flips the AP site out of the DNA helix

and into its active site. The nucleosomal DNA sculpting by APE1 occurs without a direct interaction with histones or large structural rearrangements of the histone octamer, when the AP-site is in a solvent-exposed position. The mode of DNA sculpting and AP site recognition are similar to that previously observed for APE1 for non-nucleosomal DNA[33,34], consistent with APE1 using a common mechanism for processing solvent-exposed AP sites in the nucleosome and AP sites in non-nucleosomal DNA. Interestingly, similar nucleosomal DNA distortion was recently observed during nucleosome engagement by multiple pioneer transcription factors[40,41], indicating this may be a common mechanism for accessing binding sites within nucleosomal DNA.

In contrast to the solvent-exposed AP site, the APE1 cleavage rates at occluded AP sites near SHL$_{-6.5}$ and SHL$_0$ are significantly lower,

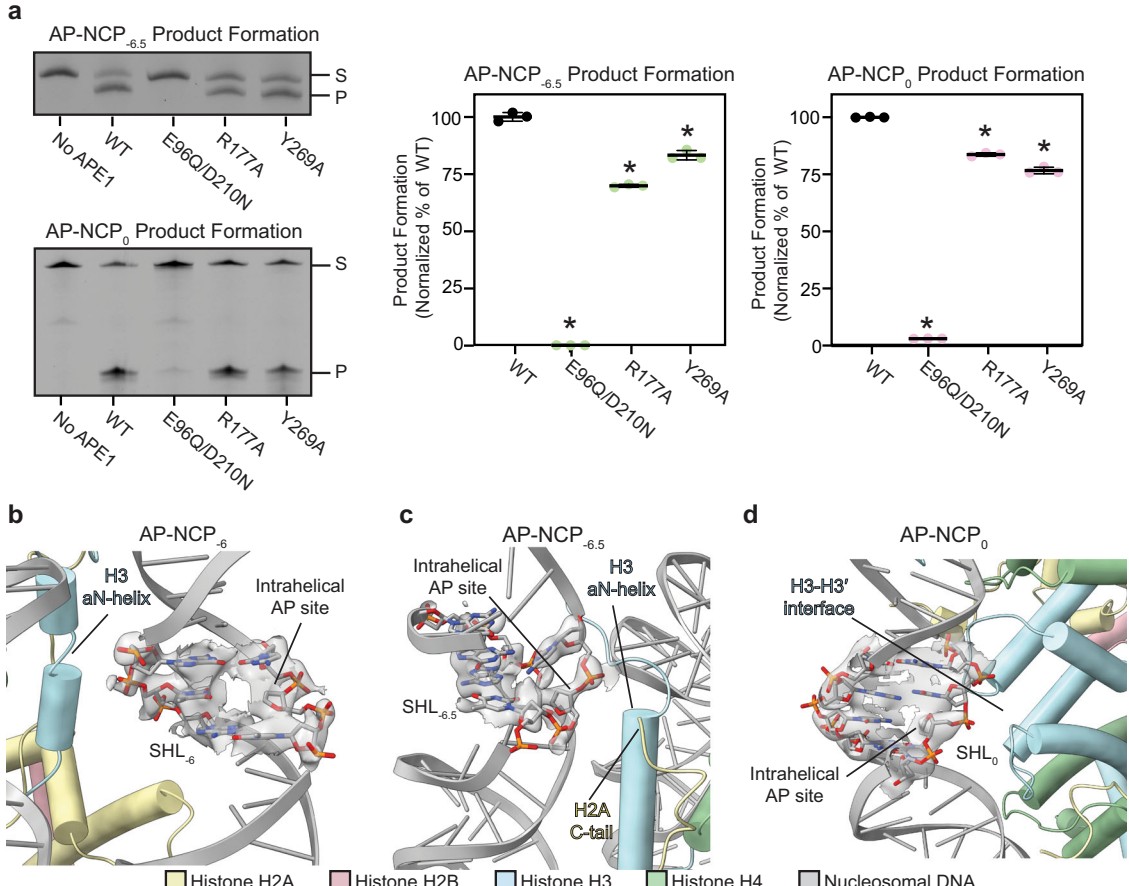

**Fig. 5 | Mechanism for APE1 processing occluded AP sites in the nucleosome.**
**a** Representative gels of the AP-NCP$_{-6.5}$ and AP-NCP$_0$ product formation assays for WT, E96Q/D210N, R177A, and Y269A APE1 (left). The substrate (S) and product (P) bands were detected using the 6-FAM label. Quantification of the AP-NCP$_{-6.5}$ and AP-NCP$_0$ product formation assays for WT, E96Q/D210N, R177A, and Y269A APE1 (right). The data shown are the mean ± standard deviation from the three replicate experiments. The * denotes values that are significantly different ($p < 0.01$) than WT APE1 as determined by two-tailed student $t$ test. $P$ values (AP-NCP$_{-6.5}$): WT vs E96Q/D210N ($p = 0.0001$), WT vs R177A (p-0.0019), and WT vs. Y269A (p-0.0086). $P$ values (AP-NCP$_0$): WT vs E96Q/D210N (p-1.7×10$^{-6}$), WT vs R177A (p-0.0005), and WT vs. Y269A (p-0.0014). **b** Focused view of the AP site in AP-NCP$_{-6}$. **c** Focused view of the AP site in AP-NCP$_{-6.5}$. **d** Focused view of the AP site in AP-NCP$_0$. The cryo-EM map surrounding the AP site is shown as a transparent gray surface in (**b**–**d**). Source data for this figure are provided as a Source Data file.

which is due to extensive interactions between the nucleosomal DNA and histone octamer that prevent efficient AP site binding, DNA sculpting, and catalysis by APE1. In this way, the histone octamer can serve as a barrier to DNA repair at the occluded AP site positions consistent with genome-wide BER profiles[28–30]. While the details remain unknown, it is likely that both intrinsic nucleosome dynamics that enhance access to the DNA damage and/or extrinsic protein factors that alter the structure of the nucleosome are required for APE1 to overcome this barrier. At SHL$_{-6.5}$, APE1 processes the AP site with a cleavage rate of -0.1 s$^{-1}$, which is almost identical to the equilibrium rate constant (0.02 - 0.1 s$^{-1}$) for spontaneous unwrapping of the nucleosomal DNA at the entry/exit site[42]. Consistent with this finding, the histone PTM H3K56ac is known to increase spontaneous unwrapping of the nucleosomal DNA and was previously shown to enhance APE1 cleavage rate at an occluded nucleosomal AP site[15]. This suggests that intrinsic nucleosome dynamics likely facilitate repair of AP sites at occluded positions near the nucleosome entry/exit through a site-exposure model.

The AP site near the nucleosome dyad (SHL$_0$) requires more extensive changes in nucleosome structure for efficient DNA repair. The UV-damaged DNA binding (UV-DDB) damage sensor protein, a recently identified BER co-factor[43], uses a register shifting mechanism to alter the conformation of DNA damage from occluded to solvent-exposed positions within the nucleosome[44]. Additional BER co-factors PARP1 and the scaffolding protein XRCC1 are also known

to directly and/or indirectly regulate nucleosome structure[24,45,46]. Finally, a variety of ATP-dependent chromatin remodeling complexes have been shown to enhance the in vitro activity of BER proteins and have been implicated in the cellular repair of DNA base damage[30,47–51]. These BER co-factors and chromatin remodeling enzymes would facilitate access of APE1 to occluded DNA damage in the nucleosome and enhance BER. However, the complex interplay between core BER factors, regulatory BER co-factors, and chromatin remodelers during DNA repair in the nucleosome remains poorly understood.

In addition to APE1, multiple BER proteins including DNA glycosylases and DNA polymerase β bend non-nucleosomal DNA ~30° to 90° during damage recognition within non-nucleosomal DNA[13,14] Our observation that APE1 adopts a similar conformation during AP site recognition in nucleosomal DNA and non-nucleosomal DNA indicates that DNA sculpting is likely needed for efficient repair in the nucleosome[33,34]. Whether DNA glycosylases and DNA polymerase β also sculpt nucleosomal DNA during damage recognition remains unknown, but the use of a DNA sculpting mechanism similar to that observed for APE1 would explain their position-dependent activity in the nucleosome[18,20–22,26,52–56]. Future work will be needed to identify the structural basis for DNA damage recognition by other core BER factors and whether these enzymes use a unified DNA sculpting mechanism for processing DNA damage in the nucleosome.

## Methods

### Purification of full-length APE1

Codon optimized *H. sapien* wild-type full-length (FL) APE1 in a pet28a vector was purchased from GenScript. The E96A/D210N, R177A, Y269A APE1 mutants were generated using the QuikChange II site-directed mutagenesis kit (Agilent). All proteins were expressed in One Shot BL21(DE3) plysS E. coli cells (Invitrogen), grown at 37 °C to an $OD_{600}$ = 0.6, and induced with 0.4 mM IPTG overnight at 20 °C. Cells were subsequently harvested and lysed via sonication on ice in a buffer containing 50 mM HEPES (pH 7.4), 50 mM NaCl, and a protease inhibitor cocktail (AEBSF, leupeptin, benzamidine, pepstatin A). The cell lysate was cleared for 1 h at 24,242 × *g*. The supernatant containing APE1 was purified via a HiTrap Heparin HP (GE Health Sciences) equilibrated with 50 mM HEPES (pH 7.4) and 50 mM NaCl, and APE1 eluted off the column with a linear salt gradient with 50 mM HEPES (pH 7.4) and 1 M NaCl. The eluted APE1 was diluted to 50 mM NaCl and further purified by cation-exchange chromatography using a POROS HS column (GE Health Sciences). APE1 was eluted from the POROS HS column with a linear salt gradient to 1 M NaCl. APE1 protein was further purified by gel filtration on a HiPrep 16/60 Sephacryl S-200 HR (GE Health Sciences). The purified APE1 protein was concentrated to >20 mg/mL and stored long term at −80 °C. All APE1 concentrations were determined via UV–Vis spectroscopy using a NanoDrop One Spectrophotometer (Thermo Scientific).

### Preparation of oligonucleotides

To generate DNA substrates containing the 601 strong positioning sequence and single AP sites, a ligation-based method was utilized[56]. The individual DNA oligonucleotides (oligos) were synthesized by Integrated DNA Technologies (Supplementary Table 2). Each set of oligos were resuspended in a buffer containing 10 mM Tris-pH 8.0 and 1 mM EDTA and mixed at a 1:1 ratio. The oligos were annealed by heating to 90 °C for 5 min before stepwise cooling to 4 °C using a linear gradient (-1 °C/min). The annealed DNA was subsequently ligated with T4 DNA ligase (CapeBio or New England Biolabs) and products separated via denaturing urea polyacrylamide gel electrophoresis (10% 37.5:1 acrylamide:bis-acrylamide). The ligated DNA was extracted from the gel in a buffer containing 200 mM NaCl and 1 mM EDTA using freeze-thaw cycles (3x, -20 °C to 37 °C). The purified and ligated DNA was reannealed heating to 90 °C for 5 min before stepwise cooling to 4 °C using a linear gradient (-1 °C/min). All ligated oligonucleotides were stored long term at −20 °C.

### Purification of recombinant histones

Recombinant *H. sapien* histone H2A (UniProt identifier: P0C0S8), *H. sapien* H2B (UniProt identifier: P62807), *H. sapien* H3 C110A (UniProt identifier Q71DI3), and *H. sapien* H4 (Uniprot identifier: P62805) proteins were generated from a tagless pet3a expression vector. In brief, each histone plasmid was transformed and expressed in BL21 (DE3) (New England Biolabs) or BL21-CodonPlus (Agilent). Cells were grown in M9 minimal media at 37 °C until an $OD_{600}$ of 0.4 was reached. Histone expression was induced with 0.3 mM (Histone H4) or 0.4 mM (Histone H2A, H2B, and H3) IPTG for 3-4 h at 37 °C. Cell pellets were stored at −80 °C. Histone purification was performed using a previously established protocol[57]. In short, histones were extracted from inclusion bodies under denaturing conditions. Following extraction, the histones were further purified using a combination of anion-exchange and cation-exchange chromatography. The purified histones were dialyzed into $H_2O$, lyophilized, and stored long term at −20 °C.

### Refolding of H2A/H2B Dimers and H3/H4 Tetramers

To generate H2A/H2B dimers and H3/H4 tetramers, each individual histone was resuspended in a buffer containing 20 mM Tris (pH 7.5), 6 M Guanidinium-HCl, and 10 mM DTT. The appropriate histones were mixed 1:1 molar ratio and dialyzed three time against a high salt buffer

containing 2 M NaCl, 20 mM Tris (pH-7.5), and 1 mM EDTA. The refolded H2A/H2B dimers and H3/H4 tetramers were purified over a Sephacryl S-200 HR (GE Health Sciences) using high salt buffer containing 2 M NaCl, 20 mM Tris (pH-7.5), and 1 mM EDTA. The fractions containing pure H2A/H2B dimer and H3/H4 tetramer were combined and stored long term in 50% glycerol at −20 °C.

### Preparation of nucleosomes

All nucleosomes were generated via a modified salt-dialysis method[57]. In short, H2A/H2B dimer, H3/H4 tetramer, and DNA were mixed at a 2:1:1 ratio, respectively. Nucleosomes were reconstituted via step-wise dialysis from 2 M NaCl to 1.5 M NaCl, 1.0 M NaCl, 0.66 M NaCl, 0.50 M NaCl, 0.25 M NaCl, and 0.125 M NaCl over 24–36 h. The reconstituted nucleosomes were heat shocked at 55 °C to obtain uniform DNA positioning prior to purification via ultracentrifugation through a 10-40% sucrose gradient. Nucleosome formation and purity was determined by running native polyacrylamide gel electrophoresis (5% 59:1 acrylamide:bis-acrylamide). The NCPs for AP-endonuclease assays and EMSAs contained a 6-FAM label at the 5′-end of the AP-site containing I strand (Supplementary Table 2).

### APE1 AP-Endonuclease Activity Assays

Single-turnover reactions were initiated by mixing full length APE1 enzyme (1000 nM) and AP-NCP substrate (100 nM) solutions in a reaction buffer containing 50 mM HEPES (pH-7.5), 100 mM KCl, 5 mM $MgCl_2$ and 0.1 mg/ml bovine serum albumin (BSA) at 37 °C. A rapid quench flow system (KinTek RQF-3) was utilized for single-turnover pre-steady-state experiments time courses for AP-NCP$_{-6}$ (2.6 ms.–15 s.), while time courses for AP-NCP$_{-6.5}$ (5 s.–600 s.) and AP-NCP$_0$ (20 s.–10 h) were completed in a benchtop heat block at 37 °C. The reactions were subsequently quenched with 300 mM EDTA at each respective time point. For mutant APE1 product formation assays, only a single time-point was taken at 0.1 s. for AP-NCP$_{-6}$, 180 s. for AP-NCP$_{-6.5}$, and 10 hr. for AP-NCP$_0$. All quenched reactions were mixed in 1:1 v/v ratio with a loading dye containing 100 mM EDTA, 80% deionized formamide, 0.25 mg/ml bromophenol blue and 0.25 mg/ml xylene cyanol. The reactions were incubated at 95 °C for 6 min and separated by 15% 29:1 denaturing polyacrylamide gel electrophoresis. The bands corresponding to substrate and product were visualized by the 6-FAM label (5′-end of the AP-site containing I strand) using an Amersham Typhoon RGB imager. The substrate and product bands were quantified using ImageQuant, and the data were best fit to the double exponential equation:

$$\text{Product} = A\left(1 - e^{-k1t}\right) + B\left(1 - e^{-k2t}\right) \tag{1}$$

Each time point represents an average of at least three independent replicate experiments ± the standard error of the mean. The indicated product formation for each APE1 mutant is the average of at least three independent replicate experiments ± the standard deviation. Uncropped gels for the single-turnover kinetics and product formation assays can be found in the Source Data file.

### Electrophoretic mobility shift assays

Samples for electrophoretic mobility shift assays (EMSAs) were generated by mixing 20 nM AP-NCP with increasing concentrations of FL APE1 protein (5-1000 nM) in a buffer containing 10 mM HEPES (7.5), 25 mM NaCl, 1 mM EDTA, 1 mM DTT, and 0.1 mg/mL BSA. For APE1 mutant EMSAs, samples were generated by mixing 20 nM AP-NCP with increasing concentrations of mutant APE1 protein (0 nM, 25 nM, and 250 nM) in a buffer containing 10 mM HEPES (pH-7.5), 25 mM NaCl, 1 mM EDTA, 1 mM DTT, and 0.1 mg/mL BSA. The EMSA samples were mixed with equal volume 10% sucrose loading dye and complexes separated by native polyacrylamide gel (5%, 59:1 acrylamide:bis-acrylamide) electrophoresis in a running buffer containing 0.2x TBE for

45 min at 4 °C. The bands corresponding to free NCP and bound complexes were visualized by the 6-FAM label (5′-end of the AP-site containing I strand) using an Amersham Typhoon RGB imager. The intensity of the bands was quantified using ImageJ[58], and the data fit to a one-site binding model accounting for ligand depletion:

$$AB = \frac{\left(A_T + B_T + K_{D,app}\right) - \sqrt{\left(A_T + B_T + K_{D,app}\right)^2 - 4\left(A_T B_T\right)}}{2} \quad (2)$$

where $A_T$ is the APE1 concentration, $B_T$ is the NCP concentration, and AB is the concentration of APE1:NCP complex. Each data point represents an average of at least three independent replicate experiments ± the standard deviation. For the APE1 mutant EMSAs, three independent replicate experiments were performed. Uncropped gels for the EMSAs can be found in the Source Data file.

### Cryo-EM sample and grid preparation

To generate APE1-AP-NCP$_{-6}$, 125 μL of AP-NCP$_{-6}$ was incubated with 125 μL of FL APE1 at a 1:1.5 molar ratio (~6 μM AP-NCP and ~9 μM APE1) in a buffer containing 25 mM HEPES (pH 7.4), 50 mM NaCl, 1 mM TCEP, and 5 mM EDTA. The APE1-AP-NCP$_{-6}$ sample was incubated for 10 min at 4 °C before the addition of 0.20% glutaraldehyde for crosslinking. The APE1-AP-NCP$_{-6}$ sample was allowed to crosslink for an additional 10 min and immediately purified by gel filtration using a Superdex S200 Increase 10/300 GL (GE Health Sciences) in a buffer containing 25 mM HEPES (pH 7.4), 50 mM NaCl, 1 mM TCEP, and 5 mM EDTA. A gel corresponding to the APE1-AP-NCP$_{-6}$ used for cryo-EM grid preparation can be found in Supplementary Fig. 2a.

The AP-NCP$_{-6}$, AP-NCP$_{-6.5}$, and AP-NCP$_0$ samples were generated during attempts to crosslink APE1, though these attempts yielded only structures of the AP-NCPs in the absence of APE1 (Supplementary Figs. 3, 6, 7). In brief, each AP-NCP was incubated with 125 μL of FL APE1 at a 1:1.5 molar ratio (~6 μM AP-NCP and ~9 μM APE1) in a buffer containing 25 mM HEPES (pH 7.4), 50 mM NaCl, 1 mM TCEP, and 5 mM EDTA. The samples were then incubated for 10 min at 4 °C before the addition of 0.20% glutaraldehyde for crosslinking. The samples were allowed to crosslink for an additional 10 min and immediately purified by gel filtration using a Superdex S200 Increase 10/300 GL (GE Health Sciences) in a buffer containing 25 mM HEPES (pH 7.4), 50 mM NaCl, 1 mM TCEP, and 5 mM EDTA. Gels corresponding to the AP-NCP$_{-6}$, AP-NCP$_{-6.5}$, and AP-NCP$_0$ samples used for cryo-EM grid preparation can be found in Supplementary Figs. 3a, 6a, 7a. Of note, a slower mobility band was present in each sample used for cryo-EM grid preparation, though we did not observe APE1 bound during 2D or 3D classification of the datasets.

All cryo-EM samples were concentrated to 0.20 mg/mL for APE1-AP-NCP$_{-6}$, 0.20 mg/mL for AP-NCP$_{-6}$, 0.25 mg/mL for AP-NCP$_{-6.5}$, and 0.25 mg/mL for AP-NCP$_0$ and subsequently stored at 4 °C until grid preparation. Cryo-EM grids for the APE1-AP-NCP$_{-6}$, AP-NCP$_{-6}$, AP-NCP$_{-6.5}$, and AP-NCP$_0$ samples were generated by applying 3 μL (0.20-0.25 mg/mL) of each individual sample to a different Quantifoil R2/2 300 mesh copper grid that was glow-discharged for 60 seconds. The grids were blotted for 1-3 seconds at 8 °C and 95% humidity before being plunge-frozen in liquid ethane using an FEI Vitrobot Mark IV.

### Cryo-EM data collection and processing

All cryo-EM data collections were performed at the Pacific Northwest Center for Cryo-EM (PNCC) using SerialEM. The datasets for 7U50, 7U51, and 7U53 were collected using an FEI Titan Krios 300 kV cryo-electron microscope equipped with a Gatan K3 direct electron detector (PNCC Krios 1, Supplementary Table 1). The dataset for 7U52 was collected using an FEI Titan Krios 300 kV cryo-electron microscope equipped with a Gatan BioQuantum K3 direct electron detector (PNCC Krios 4, Supplementary Table 1). For the APE1-AP-NCP$_{-6}$ dataset, a total

of 11,300 movies were recorded over ~48 h from two separate cryo-EM grids. The data was collected using super-resolution mode with a pixel size of 0.40075 Å, a defocus range of −0.8 μm to −2.2 μm, and a total electron dose of 50 e⁻/Å². For the AP-NCP$_{-6}$ dataset, a total of 3510 movies were recorded over ~24 h. from a single cryo-EM grid. The data was collected using super-resolution mode with a pixel size of 0.40075 Å, a defocus range of −0.8 μm to −2.2 μm, and a total electron dose of 50 e⁻/Å². For the AP-NCP$_{-6.5}$ dataset, a total of 3,680 movies were recorded over a ~24 h. from a single cryo-EM grid. The data was collected using super-resolution mode with a pixel size of 0.415 Å, a defocus range of -0.7 μm to −2.1 μm, and a total electron dose of 47 e⁻/Å². For the AP-NCP$_0$ dataset, a total of 4,995 movies were recorded over a single day from a single cryo-EM grid. The data was collected using super-resolution mode with a pixel size of 0.40075 Å, a defocus range of −0.8 μm to −2.0 μm, and a total electron dose of 50 e⁻/Å².

All cryo-EM data processing was carried out using cryoSPARC[59], and similar schemes were used to obtain each of the four cryo-EM structures. A correction for beam-induced drift was carried out using cryoSPARC patch motion correction and contrast transfer function (CTF) fit using cryoSPARC patch CTF-estimation. The micrographs were then manually curated. A random subset (~500) of the manually curated micrographs were used to perform blob picking to generate templates for automated template picking. After template picking, a minimum of two rounds of 2D-classification were carried out to generate the final particle stacks. Ab-initio models were generated from the final particle stacks and multiple rounds of cryoSPARC heterogenous refinement were performed. The reconstructions for AP-NCP$_{-6}$, AP-NCP$_{-6.5}$, and AP-NCP$_0$ were obtained after a final non-uniform refinement. After heterogenous refinement, a cryoSPARC 3D classification step was performed on the 294,348 particles of APE1-AP-NCP$_{-6}$ complex using a mask for APE1 and the ~15 bps of nucleosomal DNA composed of the APE1 binding site region. The class containing the most homogenous APE1-AP-NCP$_{-6}$ complex was then subjected to a non-uniform refinement step to generate the final reconstruction. To resolve the conformational heterogeneity in the APE1-AP-NCP$_{-6}$ complex, a three component cryoSPARC 3D variability analysis[60] was performed on the initial 294,348 particles of the APE1-AP-NCP$_{-6}$ complex. Further details for the cryo-EM processing pipeline for each of the four reconstructions can be found in Supplementary Figs. 2, 3, 6, 7.

The global resolution for all four structures was determined using Fourier shell correlation (FSC) 0.143 cut off. The AP-NCP$_{-6}$, AP-NCP$_{-6.5}$, and AP-NCP$_0$ reconstructions were further subjected to B-factor sharpening using PHENIX[61] autosharpen. For the APE1-AP-NCP$_{-6}$ complex, the reconstruction was subjected to two separate B-factor sharpening steps. The first sharpening step was performed using the local resolution of the nucleosome and the second sharpening performed to the local resolution of the APE1/DNA region using PHENIX autosharpen[61]. The sharpened maps were then combined using PHENIX[61] combine focused maps to yield the final composite map of the APE1-AP-NCP$_{-6}$ complex.

### Model building and refinement

A human nucleosome model was generated using the coordinates from a yeast nucleosome structure (PDB:4JJN), which was chosen due to the strong similarity in the 601 positioning DNA sequence used in our study. The yeast histone residues were then mutated to the corresponding human histone residues using Coot[62]. The initial APE1 model was generated from a high-resolution X-ray crystal structure of an APE1-AP-DNA complex (PDB:5DFI)[33]. The initial models of the human nucleosome and APE1 were rigid-body docked into the APE1-AP-NCP$_{-6}$ map using University of California San Francisco (UCSF) Chimera[63]. For AP-NCP$_{-6}$, AP-NCP$_{-6.5}$, and AP-NCP$_0$, the initial model of the human nucleosome was rigid-body docked into the respective cryo-EM maps using UCSF Chimera[63]. All models were iteratively

refined in PHENIX[61] using secondary structure restraints and manual adjustments made to side chain conformations using Coot[62]. MolProbity[64] was used to validate the final models prior to deposition. The model coordinates for the APE1-AP-NCP$_{-6}$, AP-NCP$_{-6}$, AP-NCP$_{-6.5}$, and AP-NCP$_0$ were deposited into the protein data bank under accession numbers 7U50, 7U51, 7U52, and 7U53. The cryo-EM maps for the APE1-AP-NCP$_{-6}$, AP-NCP$_{-6}$, AP-NCP$_{-6.5}$, and AP-NCP$_0$ were deposited into the electron microscopy data bank under accession numbers EMD-26336, EMD-26337, EMD-26338, and EMD-26339. All figures of the cryo-EM maps and models were generated using UCSF ChimeraX[65].

### Reporting summary
Further information on research design is available in the Nature Research Reporting Summary linked to this article.

## Data availability
Atomic coordinates for the reported structures have been deposited with the Protein Data bank under accession numbers 7U50, 7U51, 7U52, and 7U53. All cryo-EM maps are available from the electron microscopy data bank under accession numbers EMD-26336, EMD-26337, EMD-26338, and EMD-26339. Atomic coordinates for the initial nucleosome model were obtained from the Protein Data bank under accession number 4JJN. Atomic coordinates for the APE1-AP-DNA structure were obtained from the Protein Data bank under accession number 5DFI. The enzyme kinetics and EMSAs generated in this study are available in the Supplementary Information file and the source data file. Source data are provided with this paper.

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

## Acknowledgements

We thank Todd Washington (University of Iowa), Amy Whitaker (Fox Chase Cancer Center), and Alexandra Machen (University of Kansas Medical Center) for helpful discussion and assistance with the manuscript. We also thank Nancy Meyer for assistance with cryo-EM screening and data collection at PNCC. This research was supported by the National Institute of General Medical Science R35-GM128562 and R01-ES029203 (B.D.F., T.M.W., N.M.H., and J.J.S.) and F32-GM140718 (T.M.W.). A portion of this research was also supported by NIH grant U24GM129547 and performed at the PNCC at OHSU and accessed through EMSL (grid.436923.9), a DOE Office of Science User Facility sponsored by the Office of Biological and Environmental Research. The research reported in this publication was also supported by the National Cancer Institute Cancer Center Support Grant P30 CA168524. Initial cryo-EM samples were prepared and imaged with assistance from Ed Brignole at the Automated Cryogenic Electron Microscopy Facility in MIT.nano on a Talos Arctica microscope, which was a gift from the Arnold and Mabel Beckman Foundation.

## Author contributions

T.M.W. and B.D.F. conceptualized the experiments. T.M.W., N.M.H., and J.J.S. generated/purified the nucleosomes and APE1 proteins for enzyme kinetics, EMSAs, and cryo-EM experiments. N.M.H. performed the enzyme kinetics. T.M.W. and J.J.S performed the EMSAs. T.M.W., N.M.H., J.J.S., and B.D.F. analyzed the enzyme kinetics and EMSAs. T.M.W., L.G., and N.J.S. performed cryo-EM sample preparation. T.M.W., N.J.S., and B.D.F processed and analyzed the cryo-EM datasets. T.M.W., N.J.S., and B.D.F performed model building and refinement. T.M.W., N.M.H., and B.D.F wrote the manuscript with input from J.J.S., L.G., and N.J.S.

## Competing interests

The authors declare no competing interests.
