## [Peer Review File · Nature Communications]

Structural basis for APE1 processing DNA damage in the nucleosomeREVIEWER COMMENTS

Reviewer #1 (Remarks to the Author):

This manuscript provides new insights into how APE1, a key factor in base excision repair (BER), recognizes nucleosomal AP sites and exerts its enzymatic activity. The authors generated and analyzed three NCP containing AP sites positioned at Dyad (NCP0), SHL-6 (NCP-6), and SHL-6.5 (NCP6.5), respectively. The processing activity of APE1 against these AP sites was analyzed, and the authors found that APE1 processes the AP site near SHL-6 with a similar rate of cleavage to that of non-nucleosomal DNA. In contrast, for NCP-6.5, containing an AP site positioned on the side facing the histones, and for NCP0, the authors found that the cleavage activity of APE1 is reduced compared to NCP-6. The authors speculated that the differences measured in APE1 cleavage activity may be due to the accessibility of APE1 to different AP sites on nucleosomes. The authors successfully determined the structure of the APE1-AP-NCP-6 complex by cryo-electron-microscopy in which APE1 binds to NCP-6. Structural analysis revealed that APE1 binds to nucleosomal DNA containing the AP site in a similar mode to that of non-nucleosomal DNA, without generating contact with histones. The authors also observed that APE1 binding slightly pulled the nucleosomal DNA outward, facilitating APE1 binding to the substrate. The authors also showed that the interaction between APE1 and the nucleosome is stabilized by three distinct interfaces, at SHL-5.5 and SHL-6.5, in addition to the APE1 active site. By generating mutants of APE1 active site, the authors showed that several amino acids in these interfaces affect the cleavage activity of APE1.

Although the manuscript contains interesting observations and is well written, the core data on the structural analysis of APE1-AP-NCP-6 give an impression of redundancy with respect to previously reported AP site-containing nucleosome structures and DNA-APE1 complex structures. It is somewhat unexpected that the authors did not find characteristic features of nucleosomes, such as the interaction between histones and APE1. The data are too weak in novelty and impact making the manuscript not suitable for Nat. Commun.

Major comments:

- 1) Now that the APE1 binding structure to NCP-6 has been determined, it will be interesting to design the next AP at a position shifted by a few bases away from SHL-6 and analyze its effect on APE1 activity and binding structure. The manuscript will increase in interest by analyzing an intermediate structure of the complex.
- 2) SHL-6 may have unique properties because of its proximity to entry/exit in the nucleosome; it would be interesting to also analyze structures where AP sites are positioned at SHL-2 and SHL-3 to see if the same outward bending occurs, as well as to perform kinetic measurements of APE1.
- 3) By generating mutants of APE1 active site, the authors showed their effects on APE1 activity, but it would be even more interesting to investigate if these mutations affect the repair process at the cellular level.
- 4) Line 93-98, Figure 1d-e, Line 184-186, Extended Fig5: In binding studies using EMSA, the authors concluded that the differences in cleavage ability were not related to APE1 binding affinities, as the binding was similar regardless of the location of the AP site, and the same goes for the results of for the E96Q/D210N, R177A, and Y269A mutants. However, the EMSA results show multiple binding, as indicated by the different up-shifted bands, making it difficult to conclude that binding of APE1 to the AP site in each SHL is similar from the loss of nucleosome bands alone. The authors should evaluate the binding affinity of APE1 to the AP site in each SHL separately.
- 5) Figure 4f, 5a: The Y269A APE1 mutant show little difference in cleavage ability between the nucleosomes which does not support the authors statement "the Y269A mutant had a larger effect on AP-NCP-6.5 and AP-NCP0 compared to the solvent-exposed AP-NCP-6" (Line 211). This part should be edited to reflect a more cautious interpretation.
- 6) Extended Figure 5b, d, f: The binding patterns of APE1 mutants appear to be different, but the text (line 184) states "the E96Q/D210N, R177A, and Y269A mutants all maintain the ability to bind AP-NCP-6, indicating the reduction in product formation is not due to reduced nucleosome binding.". The

authors should discuss more specifically the changes in the binding patterns of APE1 mutants.

7) Line 113-117: It would be very interesting to analyze the potential interactions between APE1 and the N-terminal region of histones and to include it in the current structural analysis, as well as to investigate whether APE1 contributes to nucleosomal DNA dissociation.

8) Extended Figure 2a: The authors should define "X" in lane2 of the Native-PAGE.

9) Extended Figure 3d: The range of local resolution is 3-6 Å but is almost completely red indicating that the range of resolution is not appropriate.

10) In the Methods section, the camera used is specified as Gatan K3 Summit, but it should be either Gatan K2 Summit or K3.

11) If the magnification indicated in Table 1 of the Extended data is correct, all cryo-EM data except 7U52 should have been taken by a camera other than the Gatan camera.

Minor Comments

1) Extended Table 2: The authors should indicate which oligo was used for which experiment.

2) Figure 1c, e: Error bars are missing.

3) The first gel presented in Extended Figure 1c, d, e looks the same as the gel in Figure 1b.

4) The reaction time in Extended Figure 1g is written as XX sec~.

5) The authors should provide a description for asterisks in Extended Figure 1g,3a,6a,7a.

6) Figure 4a: The authors should provide a definition for "apo" in apo-AP-NCP-6.

7) Some typos:

Line 438: cur -> cut

Figure 4d legend: theAPE1 -> the APE1

Extended Figure 1 legend: the authors should unify whether lowercase letters are enclosed in parentheses or not.

Extended Figure 5g,6g legend: g) -> g.

8) Page 13, line 273: is the term "in-vitro" correct?

Reviewer #2 (Remarks to the Author):

The authors have examined how the APE1 base excision repair endonuclease acts on a DNA substrate incorporated into a nucleosome using structural and biochemical methods. Their cryo-EM structure for APE1 bound to a solvent-exposed (outward facing) DNA site near the nucleosome entry/exit site shows that the enzyme displaces DNA slightly from the histone octamer, enough for the enzyme to bind to its target site. There are essentially no changes in structure of the histone octamer. Distortions in the DNA are similar to those observed in structures of naked DNA, i.e., not nucleosomal DNA, consistent with similar activities of the enzyme on naked vs nucleosomal substrates for this solvent-exposed target site. Unfortunately attempts to determine structures of APE1 bound to inward facing sites nucleosomes were not successful. The insights from this study should be of interest to the DNA repair field and the work is generally of high quality. I therefore favor publishing this manuscript. I also believe the manuscript will regrettably attract less attention in the chromatin field given the relative lack of new information regarding how chromatin proteins bind to the nucleosome.

Some specific comments that should be addressed in a revised manuscript:

1. The quantitative analysis of the electrophoretic mobility shift assays (EMSA) is simplistic. The authors note that they fit the data to a one-site binding model even the gels show more than a single shifted species. I am particularly concerned by what I interpret as non-specific binding of APE1 to the nucleosome. I am uncomfortable with determining apparent dissociation constants and the comparison between nucleosome substrates based on this single parameter, for example the statement that APE1 has a ~3.5-fold higher affinity for AP-NCPs (lines 100-101) versus non-damaged nucleosomes. I appreciate that rigorous analysis of their EMSA data is challenging but the authors should minimally explain to the reader the limitations of their current quantitative analysis.

2. (line 101) Please also note that the data for the ND-NCP EMSA is shown in Extended Data Fig. 1k, not 3l.
3. (p. 7) The authors should briefly discuss how the contacts made by APE1 residues to nucleosomal DNA compare to contacts made to naked DNA observed in previous structures. Many of the contacts are similar or the same as observed in those previous structures.
4. (p. 10) The authors state they have successfully generated APE1-AP-NCP(-6.5) and APE1-AP-NCP(0) but I am not convinced based the gels shown (Extended Data Fig. 6a and 7a). What is the band labeled with an asterisk in these gels? It is not clear to me if this sample contained APE1 or not. If the samples did contain APE1 (as is suggested by line 398), what evidence exists that the band labeled with an asterisk contains APE1 aside the fact that there is a similar band in Extended Data Fig. 2a?
5. (Extended Data Fig. 2a) What does lane X correspond to?
6. (Extended Data Fig. 2a, 3a, 6a, 7a) Please label the lower band as AP-NCP with an appropriate subscript for the abasic site location, i.e., include the hyphen between AP and NCP. Leaving out the hyphen is potentially confusing to the reader when the hyphen is included apparently everywhere else in the manuscript.
7. (lines 247-249) This statement should be qualified to indicate it is true only when the AP site is solvent-exposed.
8. (Methods, line 361) Please state where the FAM label was incorporated into the nucleosome.
9. (Fig. 1 and Extended Data Fig. 5) Please state the gel detection method in the figure legend. I presume the activity assays use the FAM label while the EMSAs use ethidium bromide.

Reviewer #3 (Remarks to the Author):

The manuscript from Bret Freudenthal's group presents a cryo-EM structure of the base excision repair endonuclease APE1 bound to an apurinic/aprimidinic (AP) damaged site in a nucleosome. This is a long awaited study that illuminates how this DNA repair protein recognizes the DNA damage in the context of a nucleosome and is of immediate interest to DNA damage and repair scientific community.

In this study the authors demonstrate that the APE1's cleavage efficiency at AP-sites strongly depends on their position within the nucleosome. Remarkably, APE1 cleaves a solvent-exposed AP site within the nucleosome with the same efficiency as in non-nucleosomal DNA duplex, but is up to 3,400,000-fold less efficient on the inward facing AP site. Next, the authors were able to reconstruct the structure of APE1 bound to the solvent exposed AP-site within the nucleosome and compare it with a structure of an AP-damaged nucleosome in the absence of APE1. The authors observe a pronounced bending of the nucleosomal DNA upon APE1 binding and the displacement of the AP site from the intrahelical position within the DNA to the active site of the APE1 enzyme.

The manuscript is well written and the data are clearly presented. The cryo EM maps are of good quality and the models are well fitted to the experimental data. I recommend that the manuscript should be published as is.

RESPONSE TO REVIEWER COMMENTS

Reviewer #1 (Remarks to the Author):

This manuscript provides new insights into how APE1, a key factor in base excision repair (BER), recognizes nucleosomal AP sites and exerts its enzymatic activity. The authors generated and analyzed three NCP containing AP sites positioned at Dyad (NCP0), SHL-6 (NCP-6), and SHL-6.5 (NCP6.5), respectively. The processing activity of APE1 against these AP sites was analyzed, and the authors found that APE1 processes the AP site near SHL-6 with a similar rate of cleavage to that of non-nucleosomal DNA. In contrast, for NCP-6.5, containing an AP site positioned on the side facing the histones, and for NCP0, the authors found that the cleavage activity of APE1 is reduced compared to NCP-6. The authors speculated that the differences measured in APE1 cleavage activity may be due to the accessibility of APE1 to different AP sites on nucleosomes. The authors successfully determined the structure of the APE1-AP-NCP-6 complex by cryo-electron-microscopy in which APE1 binds to NCP-6. Structural analysis revealed that APE1 binds to nucleosomal DNA containing the AP site in a similar mode to that of non-nucleosomal DNA, without generating contact with histones. The authors also observed that APE1 binding slightly pulled the nucleosomal DNA outward, facilitating APE1 binding to the substrate. The authors also showed that the interaction between APE1 and the nucleosome is stabilized by three distinct interfaces, at SHL-5.5 and SHL-6.5, in addition to the APE1 active site. By generating mutants of APE1 active site, the authors showed that several amino acids in these interfaces affect the cleavage activity of APE1. Although the manuscript contains interesting observations and is well written, the core data on the structural analysis of APE1-AP-NCP-6 give an impression of redundancy with respect to previously reported AP site-containing nucleosome structures and DNA-APE1 complex structures. It is somewhat unexpected that the authors did not find characteristic features of nucleosomes, such as the interaction between histones and APE1. The data are too weak in novelty and impact making the manuscript not suitable for Nat. Commun.

We thank the reviewer for their careful consideration of our manuscript. However, we strongly disagree that the manuscript lacks novelty and impact, which is consistent with the consensus from reviewer 2 who stated, "I therefore favor publishing this manuscript." and reviewer 3 who stated, "I recommend that the manuscript should be published as is." Our manuscript reports the first comprehensive structural basis for DNA damage recognition and processing by a base excision repair protein in a NCP. The mechanisms BER enzymes use to recognize and process DNA damage in the context of the nucleosome has been sought after in the DNA repair field for almost 20 years, and our study provides the first structural insight into nucleosomal BER. We strongly feel this manuscript will also provide a foundation for future work aimed at understanding how the 15+ additional base excision repair proteins repair DNA damage in the nucleosome. For these reasons we agree with Reviewer 2 and 3 that this manuscript is suitable for publication in *Nature Communications*, which is further highlighted by multiple studies published in *Nature Communications* of similar impact aimed at understanding how proteins interact with the nucleosome to function in chromatin¹⁻⁶.

While we agree that several of the experimental suggestions by the reviewer are interesting, they are generally exploratory in nature, presumably to address the reviewer's perceived lack of "novelty and impact." However, they do not address core flaws within the work currently presented in this manuscript. In addition, the sheer depth of the experimental suggestions would take multiple years at a large monetary cost with no guarantee of success. Nevertheless, we address each of these individual suggestions below.

Major comments:

We have changed the “Extended Data Figs. and Tables.” to “Supplementary Figs. and Tables.” to comply with *Nature Communications* formatting. We refer to these figures as “Supplementary Figs. and Tables.” throughout the response to reviewer.

1) Now that the APE1 binding structure to NCP-6 has been determined, it will be interesting to design the next AP at a position shifted by a few bases as SHL-6 and analyze its effect on APE1 activity and binding structure. The manuscript will increase in interest by analyzing an intermediate structure of the complex.

We agree with the reviewer that further understanding the structural basis for the position-dependent activity of APE1 in the NCP is of interest. In fact, our study was designed to test exactly this by using three separate AP-sites locations in the nucleosome that represent different rotational and translational registries. As noted in the manuscript, our attempts at determining the structure of APE1 bound to AP-NCP_{.6.5}, which represents a 5 bp difference in nucleosome rotational position was unsuccessful. Attempting to determine additional structures at multiple different rotational positions (i.e. “intermediate positions”) would take significant effort for sample preparation, data collection, and data processing. In addition, this analysis would take well over a year of with no guarantee of success. Therefore, we strongly feel this experimental request is unreasonable and beyond the scope of the current manuscript. Our group will continue to work towards understanding how rotational position of the AP-site alters APE1 activity and the structural basis for this regulation, but this will require future dedicated studies, additional proposals at national cryo-EM centers, and additional financial support.

2) SHL-6 may have unique properties because of its proximity to entry/exit in the nucleosome; it would be interesting to also analyze structures where AP sites are positioned at SHL-2 and SHL-3 to see if the same outward bending occurs, as well as to perform kinetic measurements of APE1.

We agree with the reviewer that understanding how APE1 processes solvent-exposed AP-sites in different nucleosome translational registries is of interest. As we described above, attempting to determine additional structures of APE1 bound to solvent-exposed AP-sites at different translational positions in the nucleosome would require an additional year of sample preparation, data collection, and data processing, which is well beyond the time frame allowed for this manuscript revision. We strongly feel this experimental request is beyond the scope of the current manuscript.

3) By generating mutants of APE1 active site, the authors showed their effects on APE1 activity, but it would be even more interesting to investigate if these mutations affect the repair process at the cellular level.

We thank the reviewer for this insightful comment. Several of the APE1 active site mutations probed in our study have already been generated in cell culture models and the sensitivity to DNA damaging agents measured⁷. In McNeill et al., the authors show that inactivation of endogenous WT APE1 results in a significant decrease in cell viability when challenged with the DNA damaging agent methylmethane sulphonate (MMS), consistent with a critical role for APE1 in repairing DNA base damage during BER⁸. Complementation experiments with the APE1

D210N mutation (catalytically dead APE1) resulted in significantly reduced cell viability in the presence of MMS. Similarly, complementation experiments with the APE1 R177A mutation resulted in significantly reduced cell viability in the presence of MMS. To our knowledge, the effect of the APE1 Y269A mutation on cellular repair of DNA damage has not been tested.

It is important to note that the APE1 D210, R177, and Y269 residues regulate APE1 enzymatic functions on a variety of nucleic acid substrates. For example, E96Q/D210N and R177A regulate APE1 AP-endonuclease activity⁹⁻¹¹ and APE1 exonuclease activity¹²⁻¹⁴. Moreover, the APE1 E96Q/D210N and R177A mutations alter APE1 endonuclease activity on multiple nucleic acid substrates including RNA¹⁵, non-nucleosomal DNA^{9,11}, and nucleosomal DNA (this manuscript). Therefore, these active site mutations would impact the APE1 enzymatic function across multiple substrates in a complex way and are not true separation of function mutations. This makes determining whether the observed sensitivity to DNA damaging agents during cellular repair is due to deficient repair of nucleosomal AP-sites or one of the other multiple APE1 other functions. In addition, attempting to begin a rigorous study aimed at understanding the cellular effect of these APE1 mutations on AP-site repair specifically within the nucleosome and/or chromatin is beyond the scope of this manuscript.

4) Line 93-98, Figure 1d-e, Line 184-186, Extended Fig5: In binding studies using EMSA, the authors concluded that the differences in cleavage ability were not related to APE1 binding affinities, as the binding was similar regardless of the location of the AP site, and the same goes for the results of for the E96Q/D210N, R177A, and Y269A mutants. However, the EMSA results show multiple binding, as indicated by the different up-shifted bands, making it difficult to conclude that binding of APE1 to the AP site in each SHL is similar from the loss of nucleosome bands alone. The authors should evaluate the binding affinity of APE1 to the AP site in each SHL separately.

We do not completely understand what the reviewer is requesting. However, we will attempt to address our best interpretation of the reviewer comment and clarify aspects of the EMSA experiments that may have caused confusion.

The experiments performed in Fig. 1d were done with NCPs containing a single AP-site position at SHL₋₆ (Fig. 1d, left), a single AP-site position at SHL_{-6.5} (Fig. 1d, middle), or a single AP-site position at SHL₀ (Fig. 1d, right). Therefore, these experiments do evaluate the binding affinity of APE1 to a NCP containing a single AP-site at each SHL separately. This is also true for all the EMSA experiments in Supplementary Fig. 5. The experiments in Supplementary Fig. 5b were performed using NCPs containing a single AP-site position at SHL₋₆. The experiments in Supplementary Fig. 5d were performed using NCPs containing a single AP-site position at SHL_{-6.5}. The experiments in Supplementary Fig. 5f were performed using NCPs containing a single AP-site position at SHL₀. Therefore, these experiments do evaluate the APE1 mutants binding to a NCP containing a single AP-site at each SHL separately.

We agree with the reviewer that there are difficulties in quantifying EMSAs using the loss of nucleosome binding alone, which was also pointed out by Reviewer 2. The patterned binding in the EMSAs likely reflect a combination of specific (AP-site) and non-specific interactions with the nucleosomal DNA. APE1 is known to bind non-damaged and AP-site containing non-nucleosomal DNA with high affinity¹⁶. The high affinity binding of non-damaged DNA is thought to be important for APE1 to scan/search DNA to identify AP-sites¹⁷. We do not believe that EMSAs are well-suited to determining subtle differences in specific AP-site binding vs non-specific binding by APE1 particularly in the context of the nucleosome that has an excess of non-damaged DNA. We have edited the manuscript to clarify the limitations of the EMSA

analysis as was also suggested by Reviewer 2. This includes informing the reader that APE1 robustly engages each AP-NCP with similar affinity, but we cannot rule out subtle differences in specific AP-site engagement due to the non-specific binding of APE1, which is consistent with our EMSA assays performed on the non-damaged NCP (Supplementary Fig. 1k,i).

5) Figure 4f, 5a: The Y269A APE1 mutant show little difference in cleavage ability between the nucleosomes which does not support the authors statement "the Y269A mutant had a larger effect on AP-NCP-6.5 and AP-NCP0 compared to the solvent-exposed AP-NCP-6" (Line 211). This part should be edited to reflect a more cautious interpretation.

We agree with the reviewer that using a more cautious interpretation of this data is warranted. We have edited this sentence of the manuscript to tone down this statement (Line 208-209).

6) Extended Figure 5b, d, f: The binding patterns of APE1 mutants appear to be different, but the text (line 184) states "the E96Q/D210N, R177A, and Y269A mutants all maintain the ability to bind AP-NCP-6, indicating the reduction in product formation is not due to reduced nucleosome binding.". The authors should discuss more specifically the changes in the binding patterns of APE1 mutants.

We thank the reviewer for this insightful comment. For each APE1 mutant we observed saturation at 250 nM APE1 indicating that each APE1 mutant can still bind the nucleosome, which is consistent with the original conclusion that the reduction in product formation is not due to reduced nucleosome binding. As the reviewer noted, we did observe subtle changes in binding patterns for the three APE1 mutant proteins, and we have updated the text to inform the readers of these differences in binding patterns (Lines 186-189 and 202-204). However, we are not able to speculate what these subtle changes represent within the manuscript as to not overinterpret the EMSAs.

7) Line 113-117: It would be very interesting to analyze the potential interactions between APE1 and the N-terminal region of histones and to include it in the current structural analysis, as well as to investigate whether APE1 contributes to nucleosomal DNA dissociation.

We appreciate the reviewer's suggestion. The section of the manuscript the reviewer pointed to is in reference to the N-terminal domain of APE1, not the N-terminal region of the histones (i.e. histone tails). Importantly, we did not observe an interaction of APE1 with the histone tails in our cryo-EM structure. Though we could not resolve most of the N-terminal histone tails, this strongly suggests that APE1 is not forming a stable interaction with the histone tails. This is consistent with prior enzymology with APE1 showing that deletion of the N-terminal histone tails did not affect enzymatic activity at a limited set of DNA damage positions¹⁸. We do appreciate that the N-terminal histone tails may play an indirect role in regulating the enzymatic activity of APE1 by making the DNA damage more accessible at some locations (e.g. through histone post-translational modifications), which is likely highly dependent on the position of the DNA damage in proximity to each respective histone tail. However, this is beyond the scope of the current manuscript and would need a dedicated future manuscript to rigorously address.

We also agree with the reviewer that understanding whether APE1 can capture the nucleosomal DNA in an unwrapped conformation during repair would be of interest, particularly in the context of occluded AP-sites near the nucleosome entry/exit site. However, the best way to address this question would be through a complex single-molecule FRET assay^{19,20}. While our lab is interested in adopting this methodology to study APE1 and other BER proteins, we are not currently in position to perform these complex experiments.

8) Extended Figure 2a: The authors should define “X” in lane2 of the Native-PAGE.

We apologize for the confusion regarding the labeling of the Native PAGE gel. Lane “X” in Supplementary Fig. 2a denotes a sample that was run on the same gel, but not used for cryo-EM grid preparation. We have added a statement in the figure legend to describe Lane “X.”

9) Extended Figure 3d: The range of local resolution is 3-6 Å but is almost completely red indicating that the range of resolution is not appropriate.

We have updated the local resolution estimation in Supplementary Fig. 3d.

10) In the Methods section, the camera used is specified as Gatan K3 Summit, but it should be either Gatan K2 Summit or K3.

We apologize and have corrected the methods to specify the correct camera (lines 419-424). The datasets for 7U50, 7U51, and 7U53 were collected using an FEI Titan Krios 300 kV cryo-electron microscope equipped with a Gatan K3 direct electron detector (PNCC Krios 1). The dataset for 7U52 was collected using an FEI Titan Krios 300 kV cryo-electron microscope equipped with a Gatan BioQuantum K3 direct electron detector (PNCC Krios 4).

11) If the magnification indicated in Table 1 of the Extended data is correct, all cryo-EM data except 7U52 should have been taken by a camera other than the Gatan camera.

The magnification for the 7U52 data collection has been updated from 130,000x to 105,000x (Supplementary Table 1). We apologize for this error during the initial submission. However, the statement that the datasets for 7U50, 7U51, and 7U53 could not have been taken by a Gatan K3 camera based on the magnification of 29,000x is incorrect. The magnification for these datasets is indeed correct.

Minor Comments

1) Extended Table 2: The authors should indicate which oligo was used for which experiment.

We have indicated which oligos were used for each experimental technique in Supplementary Table 2.

2) Figure 1c, e: Error bars are missing.

We have updated Fig. 1e with error bars that were inadvertently omitted during figure preparation. Fig 1c. contains error bars, they are simply smaller than the size of the individual data points. We have added a statement in the figure legend noting this for Fig. 1c. We have also provided a Source Data file for all quantifications and associated error in the manuscript as per *Nature Communications* guidelines.

3) The first gel presented in Extended Figure 1c, d, e looks the same as the gel in Figure 1b.

We apologize for the confusion. The first gels in Supplementary Fig. 1c-e are the same as in Figure 1b. Figure 1b shows a representative gel from the triplicate single turnover experiments. Supplementary Fig.1c-e show the triplicate experiments so that the reader can directly compare

all replicate experiments to assess reproducibility. We have updated the Supplementary Fig. 1 legend with the following statement to alleviate the confusion: "The first gel of the triplicate experiments for each AP-NCP was used as the representative gel in Fig. 1b."

4) The reaction time in Extended Figure 1g is written as XX sec~.

This has been corrected.

5) The authors should provide a description for asterisks in Extended Figure 1g,3a,6a,7a.

A description for the asterisks in Supplementary Fig. 1g, 3a, 6a, and 7a have been provided.

6) Figure 4a: The authors should provide a definition for "apo" in apo-AP-NCP-6.

The word "apo" has been removed from Figure 4a and Supplementary Fig. 4c to conform with the nomenclature used throughout the rest of the manuscript.

7) Some typos:

Line 438: cur -> cut

Figure 4d legend: theAPE1 -> the APE1

Extended Figure 1 legend: the authors should unify whether lowercase letters are enclosed in parentheses or not.

Extended Figure 5g,6g legend: g) -> g.

We thank the reviewer for catching these typos, which have been corrected in the revised manuscript.

8) Page 13, line 273: is the term "in-vitro" correct?

The term *in vitro* is correct, and we apologize for the confusion. We have reworded the sentence to improve clarity (lines 271-273).

Reviewer #2 (Remarks to the Author):

The authors have examined how the APE1 base excision repair endonuclease acts on a DNA substrate incorporated into a nucleosome using structural and biochemical methods. Their cryo-EM structure for APE1 bound to a solvent-exposed (outward facing) DNA site near the nucleosome entry/exit site shows that the enzyme displaces DNA slightly from the histone octamer, enough for the enzyme to bind to its target site. There are essentially no changes in structure of the histone octamer. Distortions in the DNA are similar to those observed in structures of naked DNA, i.e., not nucleosomal DNA, consistent with similar activities of the enzyme on naked vs nucleosomal substrates for this solvent-exposed target site. Unfortunately attempts to determine structures of APE1 bound to inward facing sites nucleosomes were not successful. The insights from this study should be of interest to the DNA repair field and the work is generally of high quality. I therefore favor publishing this manuscript. I also believe the manuscript will regrettably attract less attention in the chromatin field given the relative lack of new information regarding how chromatin proteins bind to the nucleosome.

We thank the reviewer for the careful and thoughtful review of our manuscript. We have changed the “Extended Data Figs. and Tables.” to “Supplementary Figs. and Tables.” to comply with Nature Communications formatting. We refer to these figures as “Supplementary Figs and Tables.” throughout the response to reviewer.

Some specific comments that should be addressed in a revised manuscript:

1. The quantitative analysis of the electrophoretic mobility shift assays (EMSA) is simplistic. The authors note that they fit the data to a one-site binding model even the gels show more than a single shifted species. I am particularly concerned by what I interpret as non-specific binding of APE1 to the nucleosome. I am uncomfortable with determining apparent dissociation constants and the comparison between nucleosome substrates based on this single parameter, for example the statement that APE1 has a ~3.5-fold higher affinity for AP-NCPs (lines 100-101) versus non-damaged nucleosomes. I appreciate that rigorous analysis of their EMSA data is challenging but the authors should minimally explain to the reader the limitations of their current quantitative analysis.

We agree with the reviewer that the quantitative analysis of the EMSA experiments is simplistic, which was our attempt at preventing the overinterpretation of the data. In hindsight, we should have better clarified the limitations to the experiments. As noted by the reviewer, APE1 interacts with each AP-NCP with a combination of specific (AP-site) and non-specific nucleosomal DNA binding (Fig. 1d), which is further supported by the high affinity binding of APE1 to ND-NCPs (Supplementary Fig. k,l). We have addressed this reviewer’s concern by editing the EMSA section of the manuscript to clarify the limitations of the analysis. This includes informing the reader that APE1 robustly engages each AP-NCP with similar affinity, but we cannot rule out subtle differences in specific AP-site engagement using EMSA due to the non-specific binding (Lines 90-101).

We do also want to point out that prior work identified APE1 binds non-damaged and AP-site containing non-nucleosomal DNA with high affinity (<5 nM)¹⁶. The non-specific DNA binding of APE1 in the context of nucleosomal and non-nucleosomal DNA has biological implications for the scanning/search mechanisms used to identify AP-sites¹⁷. We do not believe that EMSAs are well-suited to determining differences in specific AP-site binding vs non-specific binding by

APE1 particularly in the context of the nucleosome that has an excess of non-damaged DNA. We are currently developing multiple single-molecule fluorescence techniques to better delineate between non-specific and specific (AP-site) binding to the nucleosomal DNA by APE1, but this is currently a work in progress.

2. (line 101) Please also note that the data for the ND-NCP EMSA is shown in Extended Data Fig. 1k, not 3l.

We thank the reviewer for catching this mistake. The corresponding text has been adjusted (Line 94).

3. (p. 7) The authors should briefly discuss how the contacts made by APE1 residues to nucleosomal DNA compare to contacts made to naked DNA observed in previous structures. Many of the contacts are similar or the same as observed in those previous structures.

We agree with the reviewer and have added a concluding statement in that paragraph highlighting the strong similarity in binding modes for APE1 with nucleosomal and non-nucleosomal DNA (Line 133-135).

4. (p. 10) The authors state they have successfully generated APE1-AP-NCP(-6.5) and APE1-AP-NCP(0) but I am not convinced based the gels shown (Extended Data Fig. 6a and 7a). What is the band labeled with an asterisk in these gels? It is not clear to me if this sample contained APE1 or not. If the samples did contain APE1 (as is suggested by line 398), what evidence exists that the band labeled with an asterisk contains APE1 aside the fact that there is a similar band in Extended Data Fig. 2a?

We apologize for the confusion regarding the formation of the APE1-NCP complexes at SHL_{-6.5} and SHL₀. We outlined this in the “Cryo-EM sample and grid preparation” section of the methods but regrettably did so in a manner that lacked clarity. The apo structures of AP-NCP₋₆, AP-NCP_{-6.5}, and AP-NCP₀ were determined from samples that were generated in the same manner as the APE1-AP-NCP₋₆ complex sample. In brief, each AP-NCP was incubated with APE1, followed by mild glutaraldehyde crosslinking. The complexes were then subjected to size exclusion chromatography (Superdex S200 Increase 10/300 GL) to purify the presumed complexes from unbound nucleosome and higher molecular weight species. Each of the AP-NCP structures determined in Fig. 5b-d and Supplementary Fig. 3a, 6a, and 7a were obtained from grids generated with samples that contained both APE1 and the respective AP-NCP at the time of crosslinking, but prior to purification. To improve clarity, we have rewritten that section of the methods to walk through the sample generation more thoroughly (lines 390-411).

We presumed the asterisk band in Supplementary Fig. 3a, 6a, and 7a was the formation of an APE1-NCP complex for each respective nucleosome. This was due to the similar difference in mobility as was seen in the EMSAs (Fig. 1d, Supplementary Fig. 1h-j) and the similar difference in mobility to the complex we used to successfully determine the APE1-AP-NCP₋₆ structure (Supplementary Fig. 2a). However, we have no direct evidence that APE1 is present in the slower migrating NCP band in the Native PAGE gel from the samples that were ultimately used to determine the apo structures of AP-NCP₋₆, AP-NCP_{-6.5}, and AP-NCP₀. As noted by the reviewer, this alone is not sufficient evidence that the corresponding band represents APE1 in complex with the respective AP-NCP. These samples were generated over a year ago and in most cases were used entirely for grid preparation. This prevents us from returning to those samples to probe the contents via an orthogonal approach.

We have removed the comment regarding the successful generation of the APE1-AP-NCP_{-6.5} and APE1-AP-NCP₀ complexes and instead highlighted that we attempted to determine the structure of these complexes unsuccessfully (lines 193-196). We originally included this statement for rigor and transparency in how the samples were generated prior to grid preparation. However, we do not want to mislead readers on the successful generation of those complexes without additional evidence, as noted by the reviewer. We have also included a statement in the Supplementary Fig. 3, 6, and 7 legends describing that the asterisk represents a crosslinking contaminant.

5. (Extended Data Fig. 2a) What does lane X correspond to?

We apologize for the confusion regarding the labeling of the Native PAGE gel. Lane “X” in Supplementary Fig. 2a denotes a sample that was run on the same gel, but not used for cryo-EM grid preparation. We have added a statement in the figure legend to describe Lane “X.”

6. (Extended Data Fig. 2a, 3a, 6a, 7a) Please label the lower band as AP-NCP with an appropriate subscript for the abasic site location, i.e., include the hyphen between AP and NCP. Leaving out the hyphen is potentially confusing to the reader when the hyphen is included apparently everywhere else in the manuscript.

We have relabeled the lower band to AP-NCP with the respective SHL location in Supplementary Fig. 2a, 3a, 6a, and 7a.

7. (lines 247-249) This statement should be qualified to indicate it is true only when the AP site is solvent-exposed.

We agree and have added a qualifier that the DNA sculpting does not require a direct interaction or large structural changes within the histone octamer at solvent-exposed positions (lines 243-245).

8. (Methods, line 361) Please state where the FAM label was incorporated into the nucleosome.

We thank the reviewer for noticing this omission. We have added additional text to describe the location of the 6-FAM label in the “Preparation of Nucleosomes” section of the methods (lines 347-349). In addition, we also added additional text in the “APE1 AP-endonuclease activity assays” and “Electrophoretic mobility shift assays” section of the methods when referring to how the nucleosome was detected (line 363 and line 380).

9. (Fig. 1 and Extended Data Fig. 5) Please state the gel detection method in the figure legend. I presume the activity assays use the FAM label while the EMSAs use ethidium bromide.

We have added the gel detection method throughout the main text and supplementary figure legends. Both single-turnover pre-steady state kinetics, EMSAs, and product formation assays were all detected with the 6-FAM label.

Reviewer #3 (Remarks to the Author):

The manuscript from Bret Freudenthal's group presents a cryo-EM structure of the base excision repair endonuclease APE1 bound to an apurinic/aprimidinic (AP) damaged site in a nucleosome. This is a long awaited study that illuminates how this DNA repair protein recognizes the DNA damage in the context of a nucleosome and is of immediate interest to DNA damage and repair scientific community.

In this study the authors demonstrate that the APE1's cleavage efficiency at AP-sites strongly depends on their position within the nucleosome. Remarkably, APE1 cleaves a solvent-exposed AP site within the nucleosome with the same efficiency as in non-nucleosomal DNA duplex, but is up to 3,400,000-fold less efficient on the inward facing AP site. Next, the authors were able to reconstruct the structure of APE1 bound to the solvent exposed AP-site within the nucleosome and compare it with a structure of an AP-damaged nucleosome in the absence of APE1. The authors observe a pronounced bending of the nucleosomal DNA upon APE1 binding and the displacement of the AP site from the intrahelical position within the DNA to the active site of the APE1 enzyme.

The manuscript is well written and the data are clearly presented. The cryo EM maps are of good quality and the models are well fitted to the experimental data. I recommend that the manuscript should be published as is.

We thank the reviewer for these kind comments and the enthusiasm for our manuscript.

Reviewer References

- 1 Wang, L., Chen, K. & Chen, Z. Structural basis of ALC1/CHD1L autoinhibition and the mechanism of activation by the nucleosome. *Nature communications* **12**, 1-9 (2021).
- 2 De Ioannes, P. *et al.* Structure and function of the Orc1 BAH-nucleosome complex. *Nature communications* **10**, 1-11 (2019).
- 3 Farnung, L., Vos, S. M. & Cramer, P. Structure of transcribing RNA polymerase II-nucleosome complex. *Nature communications* **9**, 1-6 (2018).
- 4 Bilokapic, S. & Halic, M. Nucleosome and ubiquitin position Set2 to methylate H3K36. *Nature communications* **10**, 1-9 (2019).
- 5 Tanaka, H. *et al.* Interaction of the pioneer transcription factor GATA3 with nucleosomes. *Nature communications* **11**, 1-10 (2020).
- 6 Park, S. H. *et al.* Cryo-EM structure of the human MLL1 core complex bound to the nucleosome. *Nature communications* **10**, 1-13 (2019).
- 7 McNeill, D. R. *et al.* Functions of the major abasic endonuclease (APE1) in cell viability and genotoxin resistance. *Mutagenesis* **35**, 27-38 (2020).
- 8 Xanthoudakis, S., Smeyne, R. J., Wallace, J. D. & Curran, T. The redox/DNA repair protein, Ref-1, is essential for early embryonic development in mice. *Proceedings of the National Academy of Sciences* **93**, 8919-8923 (1996).
- 9 Mol, C. D., Izumi, T., Mitra, S. & Tainer, J. A. DNA-bound structures and mutants reveal abasic DNA binding by APE1 DNA repair and coordination. *Nature* **403**, 451-456 (2000).
- 10 McNeill, D. R. & Wilson, D. M. A dominant-negative form of the major human abasic endonuclease enhances cellular sensitivity to laboratory and clinical DNA-damaging agents. *Molecular Cancer Research* **5**, 61-70 (2007).
- 11 Izumi, T., Schein, C. H., Oezguen, N., Feng, Y. & Braun, W. Effects of Backbone Contacts 3' to the Abasic Site on the Cleavage and the Product Binding by Human Apurinic/Apyrimidinic Endonuclease (APE1). *Biochemistry* **43**, 684-689 (2004).
- 12 Whitaker, A. M., Flynn, T. S. & Freudenthal, B. D. Molecular snapshots of APE1 proofreading mismatches and removing DNA damage. *Nature communications* **9**, 1-11 (2018).
- 13 Whitaker, A. M., Stark, W. J. & Freudenthal, B. D. Processing of oxidatively damaged DNA dirty ends by APE1. *bioRxiv* (2021).
- 14 Chou, K.-m. & Cheng, Y.-c. The exonuclease activity of human apurinic/aprimidinic endonuclease (APE1): biochemical properties and inhibition by the natural dinucleotide Gp4G. *Journal of Biological Chemistry* **278**, 18289-18296 (2003).
- 15 Kim, W.-C. *et al.* Characterization of the endoribonuclease active site of human apurinic/aprimidinic endonuclease 1. *Journal of molecular biology* **411**, 960-971 (2011).
- 16 Liu, Y. *et al.* Coordination of steps in single-nucleotide base excision repair mediated by apurinic/aprimidinic endonuclease 1 and DNA polymerase β . *Journal of Biological Chemistry* **282**, 13532-13541 (2007).
- 17 Carey, D. C. & Strauss, P. R. Human apurinic/aprimidinic endonuclease is processive. *Biochemistry* **38**, 16553-16560 (1999).

- 18 Beard, B. C., Stevenson, J. J., Wilson, S. H. & Smerdon, M. J. Base excision repair in nucleosomes lacking histone tails. *DNA Repair (Amst)* **4**, 203-209, doi:10.1016/j.dnarep.2004.09.011 (2005).
- 19 Luo, Y., North, J. A. & Poirier, M. G. Single molecule fluorescence methodologies for investigating transcription factor binding kinetics to nucleosomes and DNA. *Methods* **70**, 108-118 (2014).
- 20 Donovan, B. T., Chen, H., Jipa, C., Bai, L. & Poirier, M. G. Dissociation rate compensation mechanism for budding yeast pioneer transcription factors. *Elife* **8**, e43008 (2019).

REVIEWERS' COMMENTS

Reviewer #1 (Remarks to the Author):

The authors provided adequate responses and corrections to all our comments except those requiring additional experiments. Unfortunately, the authors did not provide the additional structural data requested, arguing that it would be time-consuming and very costly, and that such revisions would be a departure from the main purpose of the article.

The authors claim that their main finding is that they have revealed a structure in which APE1 is bound to the nucleosome. However, their newly determined structure was found to be similar to previously reported structures of AP-containing nucleosomes and structures of DNA-APE1 complexes. Therefore, we felt that this paper lacked sufficient novelty to be published in Nature Communications without the additional experiments we requested.

On the other hand, we also understand that obtaining additional structural data would take several years and that there is no guarantee of success. The next scientific questions that the authors intend to answer are also mentioned as well as their experimental plan, which is promising for future research.

Reviewer #2 (Remarks to the Author):

The authors have addressed all of my concerns and they have done this thoroughly. I am happy to support publication of this manuscript in Nature Communications.

RESPONSE TO REVIEWER COMMENTS

Reviewer #1 (Remarks to the Author):

The authors provided adequate responses and corrections to all our comments except those requiring additional experiments. Unfortunately, the authors did not provide the additional structural data requested, arguing that it would be time-consuming and very costly, and that such revisions would be a departure from the main purpose of the article.

The authors claim that their main finding is that they have revealed a structure in which APE1 is bound to the nucleosome. However, their newly determined structure was found to be similar to previously reported structures of AP-containing nucleosomes and structures of DNA-APE1 complexes. Therefore, we felt that this paper lacked sufficient novelty to be published in Nature Communications without the additional experiments we requested.

On the other hand, we also understand that obtaining additional structural data would take several years and that there is no guarantee of success. The next scientific questions that the authors intend to answer are also mentioned as well as their experimental plan, which is promising for future research.

We thank Reviewer 1 for the thorough review of our manuscript. We also appreciate the reviewer's understanding for the amount of time and money that would be necessary to determine additional APE1-bound nucleosome structures for this manuscript.

Reviewer #2 (Remarks to the Author):

The authors have addressed all of my concerns and they have done this thoroughly. I am happy to support publication of this manuscript in Nature Communications.

We thank Reviewer 2 for the thorough review of our manuscript.